# Learning enhances encoding of time and temporal surprise in mouse primary sensory cortex

Rebecca J. Rabinovich[1,2,3], Daniel D. Kato[1,2,3] & Randy M. Bruno [1,2,3,4] ✉

Primary sensory cortex has long been believed to play a straightforward role in the initial processing of sensory information. Yet, the superficial layers of cortex overall are sparsely active, even during sensory stimulation; additionally, cortical activity is influenced by other modalities, task context, reward, and behavioral state. Our study demonstrates that reinforcement learning dramatically alters representations among longitudinally imaged neurons in superficial layers of mouse primary somatosensory cortex. Learning an object detection task recruits previously unresponsive neurons, enlarging the neuronal population sensitive to touch and behavioral choice. Cortical responses decrease upon repeated stimulus presentation outside of the behavioral task. Moreover, training improves population encoding of the passage of time, and unexpected deviations in trial timing elicit even stronger responses than touches do. In conclusion, the superficial layers of sensory cortex exhibit a high degree of learning-dependent plasticity and are strongly modulated by non-sensory but behaviorally-relevant features, such as timing and surprise.

The established view of cortical processing assumes that primary sensory cortex performs basic tasks at an early stage in a sensory processing workflow, with more complex computations occurring downstream (for review see Grill-Spector & Malach[1]). However, there is mounting evidence that more "associative" processing occurs in primary sensory cortices and that activity in these regions is influenced by factors beyond simple sensation of a single modality[2–11]. Additionally, the canonical view implies robust stimulus responses by primary sensory cortex[12,13]; on the contrary, some primary sensory cortical cells, especially superficial layer neurons, often exhibit low levels of activity even during strong sensory stimulation[14–16].

A large volume of research now suggests that sensory-driven responses in primary cortical areas are modulated by learning and stimulus-reward associations[3,5–7,17–21]. In the superficial layers (layer 2/3) of primary visual cortex (V1), initially quiet cells begin responding to behaviorally relevant stimuli after pairing with reward[22], with representations becoming more selective and stable over the course of learning[20]. Meanwhile, structural changes have been found to occur in

layer 2/3 of primary somatosensory cortex[23], where certain groups of cells respond more strongly to touch after learning[21]. Yet the degree to which learning-related plasticity occurs in primary somatosensory cortex (S1), and how it manifests, remain unclear: in some cases, stimulus representations have been found to remain stable[24], with little plasticity[25].

Mismatch between expectation and sensation has also been proposed to influence activity in primary sensory cortex[26,27]. In fact, prediction, mismatch, and predictive coding in general have been theorized to play a crucial role in cortical processing, and may drive learning-related plasticity[28,29]. For instance, Keller and colleagues[26] showed that disturbances in optic flow, such as the sudden cessation of visual motion, led to increased V1 activity. However, an alternative mechanism has been proposed for this finding[30] whereby V1 responses increase whenever visual motion slows due to cells' velocity preferences, regardless of direction of motion or whether the change violated the animals' expectations (but see Keller et al. and Zmarz & Keller, who did not observe altered V1 activity during passive viewing

[1]Department of Neuroscience, Columbia University, New York, NY 10027, USA. [2]Kavli Institute for Brain Science, Columbia University, New York, NY 10027, USA. [3]Zuckerman Mind Brain Behavior Institute, Columbia University, New York, NY 10027, USA. [4]Present address: Department of Physiology, Anatomy & Genetics, University of Oxford, Oxford, UK. ✉e-mail: randy.bruno@dpag.ox.ac.uk

of the stimulus[26,31]). Meanwhile, perturbations in expected tactile flow mainly result in decreased neuronal activity in barrel cortex (the whisker-related portion of S1)[27]. Thus, despite the increasing consensus regarding the importance of predictive coding[29,32], the nature of mismatch signals in primary sensory cortices is still poorly understood.

The seemingly disparate observations across these studies may be interconnected if primary sensory cortices construct a model over the course of learning, placing stimuli, rewards, and other events into a temporal context. Various brain regions have been already implicated in time processing. For instance, compelling evidence of "time cells" has been demonstrated in the hippocampus. These cells specifically respond at given moments in a trial and have been proposed to contribute to working and episodic memory by linking events across a delay[33,34] and by representing their sequential order[35]. Similar sequence-driven coding schemes may be at play in striatum[36,37]. Some degree of time encoding was also identified in the amygdala and several association cortical areas[38]. As for sensory cortical regions, barrel cortical neurons were shown to be sensitive to temporally patterned stimuli[39]; conversely, temporally precise patterns of neuronal activity have been found to contain tactile information[40]. Similarly, putative "memory cells" in primate somatosensory cortex exhibit temporal patterns of activity during a tactile working memory task[41]. Some studies that found learning-related influences on V1 activity also noted that cells in V1 anticipated the timing of reward and punishment[3,19,42]. Yet, no study has directly investigated time coding in sensory cortex.

The current study asks: to what degree does learning impact the activity of neurons in superficial layers of primary somatosensory cortex? What aspects of the behavioral paradigm do these cells encode? What kind of mismatch signals are present in S1? We used 2-photon calcium imaging to measure cortical activity during the learning of a simple Pavlovian detection task. We found that learning a stimulus-reward association engages previously unresponsive cells in a longitudinally tracked population, leading to an enhanced representation of tactile stimuli. In addition, learning rendered cells able to encode *non-sensory* aspects of the behavioral paradigm and animals' experience, including animals' choice, the temporal progression of events, and the expectation of event timing.

## Results

To investigate learning-related effects on cortical activity, we used 2-photon calcium imaging to monitor the activity of GCaMP6f-expressing neurons in superficial layers (layer 2/3) of S1, while mice learned a Pavlovian whisker-based object-detection task. In our behavioral paradigm, stimuli were presented to water-restricted mice via a rotating wheel (Fig. 1a, top). On rewarded trials, a flat surface rotated into the whisker field, stopped and remained stationary for 2 s, and then rotated away, after which a water reward was presented (Fig. 1a, bottom; see Methods). On unrewarded trials, the wheel rotated to an empty position in which no surface contacted the whiskers, and no water was given. Thirsty mice licked at the water port in anticipation of the reward, and we quantified the level of anticipatory licking as a measure of learning. Initially, mice licked indiscriminately on both stimulus-present and stimulus-absent trials (Fig. 1b, left), but after learning the association between stimulus and reward, they licked preferentially during stimulus-present trials, in anticipation of reward (Fig. 1b, right; n = 10 mice). Individual animals' licking behavior was qualitatively similar to the average (see Supplementary Fig. 1).

The Pavlovian nature of the behavioral paradigm meant that an animal's actions did not impact the outcome of the trial (whether or not a water droplet emerged from the lick-port). Nonetheless, to quantify learning, we labeled mouse responses as follows: anticipatory licking on a stimulus-present trial was categorized as a "hit", suppression of licking on a stimulus-absent trial was a "correct rejection", licking on a stimulus-absent trial was a "false alarm", and lack of licking on a stimulus-present trial was a "miss". The former two response types

were considered "correct", and the latter two were "incorrect". Mice began to perform above chance in as few as three sessions, but we continued training them until their performance exceeded 70% (Fig. 1c) and in some cases exceeded 90%. Performance on this task fell to chance when whiskers were trimmed off (in 5 out of 5 whisker-trimmed mice).

Prior to training and 2-photon imaging (Fig. 1d), we used intrinsic signal imaging to locate the barrel cortex, where we would later record neuronal activity. After recording, we confirmed the imaging location by using the 2-photon laser to create a small lesion at the imaging site, and, with the help of vasculature patterns, identifying the corresponding location relative to the layer 4 barrels in post hoc histology (Fig. 1e).

### Conditioning, but not repeated stimulus exposure, enhances object representation

Across the neuronal population, cellular activity was variable: neurons responded at different times within the trial, both in naïve and expert sessions (example mouse in Fig. 2a, top). For most mice, average responses to the stimulus increased in amplitude as mice became proficient at the detection task: on early days, the average population activity was relatively flat; after conditioning, the activity peaked in response to the onset and offset of the stimulus (Fig. 2a, bottom; same example mouse). In addition to analyzing the overall change in fluorescence, we extracted the times of calcium transients (see Methods; Supplementary Fig. 2). All subsequent analyses are based on the time or rate of these calcium transients, unless otherwise noted. We quantified the neuronal population response for each mouse on early and expert sessions as the mean number of calcium transients across cells. Population responses to object arrival increased from early to expert days (p = 0.001, two-sided paired t-test, n = 10 mice; Fig. 2b).

Averaging signals across all cells obscures the variability in cell responses within the population, so we next examined individual cell activity. The responses of individual cells ($n_{early}$ = 1163 cells, $n_{expert}$ = 1112 cells) could be categorized into several groups based on the timing of their calcium transients (Fig. 2c; see Methods for classification criteria). Some cells responded specifically at the onset of the stimulus ("on" cells), while others responded at the stimulus offset ("off" cells); another group responded at both onset and offset ("on-off" cells); finally, a small subset only responded late in the trial, following reward presentation ("reward" cells). The "on," "off," and "on-off" cells dominated layer 2/3 S1 activity, as can be seen in the average rate of calcium transients across all imaged cells: transients occurred most frequently at stimulus onset and offset as well as, to a lesser extent, the period in between (Fig. 2d). This effect was more pronounced in expert mice, consistent with the increased population of these cells (as described below).

In addition to these four response profiles, many cells displayed no obvious response patterns ("none" cells): in fact, in naïve mice, the majority of cells were in this category (Fig. 2e, left). We refer to these neurons as unresponsive to contrast them with cells that exhibited immediate responses at the time of object arrival, departure, or reward delivery; this category may include cells that were active but did not meet our classification criteria for the other cell groups. As mice learned, the proportion of these unresponsive "none" cells decreased (p < 0.001, Z approximation to a binomial), and the proportion of stimulus-responsive cells increased (p < 0.001 for "on" cells and p < 0.001 for "off" cells). In expert mice, most cells were stimulus-responsive (Fig. 2e, right).

Was this substantial increase in the population of cells responsive to object arrival a genuine effect of conditioning, or could mere repeated exposure to stimuli suffice to increase cells' responses? We explored this issue using a new cohort of animals—mice that were exposed to the same behavioral paradigm with respect to stimulus presentation, but were not water-restricted and never received

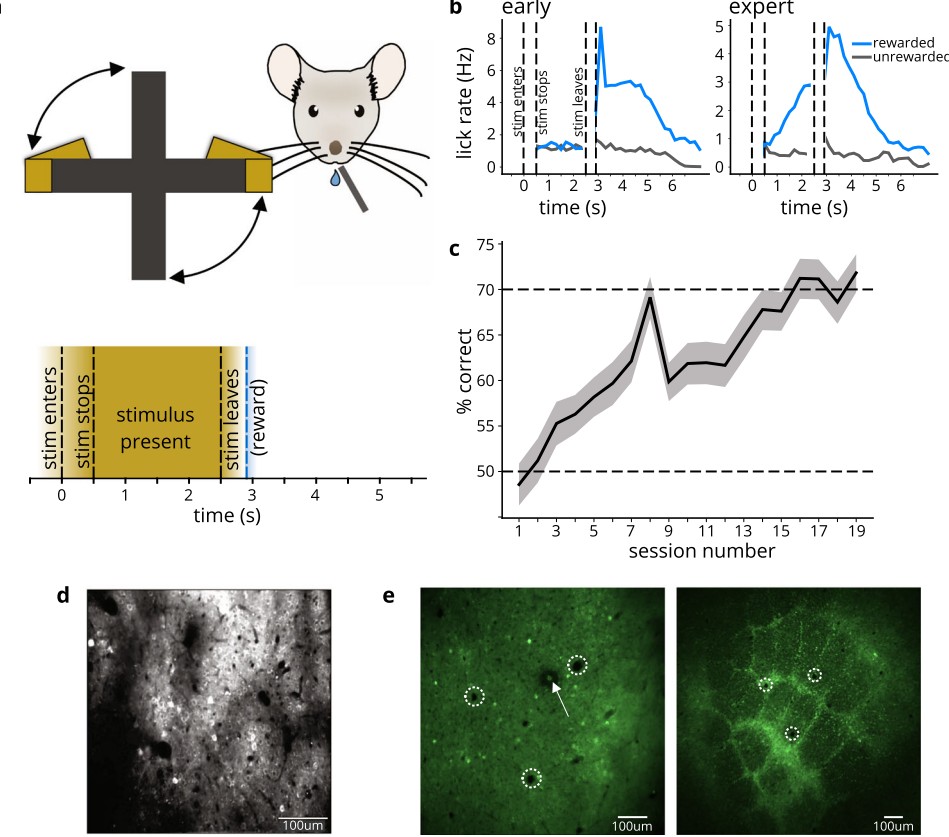

**Fig. 1 | L2/3 calcium imaging in barrel cortex while mice learn whisker-based object detection task. a** Behavior schematic (top) and timecourse (bottom): rotating wheel brings either object or empty space into mouse whisker field. Stimulus "enters" the whisker field and then "stops". The now stationary stimulus is present for 2 s before rotating away (45 degrees over the course of ~400 ms, such that the gap between wheel arms is in front of the animal's face). On object-present trials, water drop is given following the stimulus. Anticipatory licks are counted during the "stimulus present" interval. **b** Anticipatory licking histograms ($n = 10$ mice) for early (left) and expert (right) days, quantifying the rate of licking for stimulus-present rewarded (blue) trials and stimulus-absent unrewarded (black) trials. Mice initially lick equally for object-present (rewarded) and object-absent (unrewarded) conditions, but learn to lick preferentially for the object-present condition with training. The panel includes shaded areas corresponding to SEMs, but these are too small to be seen. **c** Learning curve ($n = 10$ mice). Solid line is the performance, calculated as number of correct trials for all mice, normalized by the total number of trials in each session. Shading corresponds to 95% confidence interval. Source data provided in a Source Data file. **d** Example field of view for calcium imaging during learning. Calcium activity was measured in GCaMP6f-expressing neurons in layer 2/3 of barrel cortex. **e** Intrinsic signal imaging targeting of 2-photon imaging (all mice) was confirmed by histology in 2 mice. Left: Example L2/3 tangential slice, showing lesion location (marked with arrow). Right: L4 tangential slice from same mouse, showing corresponding area of barrel map. Dashed circles indicate blood vessels used to match location in barrel map.

rewards ($n = 7$ mice). Comparing the change in proportion of cells of each type as mice progressed through the protocols revealed that the presence of reward was in fact crucial for the increase in stimulus-responsive cells. Repeated exposure produced no increase in the percentage of responsive cells; rather, we saw a decrease in the proportion of these cells ($p < 0.001$ for "on" cells; $p = 0.03$ for "off" cells), with unresponsive "none" cells growing to an even larger population ($p < 0.001$) after repeated exposure to the stimuli—the reverse of the effect seen in conditioned mice (Fig. 2f). These findings suggest that two opposing adaptation mechanisms exist in layer 2/3: irrelevant stimuli shrink the population of responsive cells while behaviorally important stimuli enlarge it.

### Learning switches response category of longitudinally tracked neurons

As mice learned the behavioral task above, a neuronal population initially dominated by unresponsive cells transformed into a population mainly comprising stimulus-responsive cells. However, that analysis was unable to reveal the dynamics of individual cells: what degree of fluctuation or stability did individual cells exhibit across consecutive days and over the course of learning? To address this question, we examined how individual cell responses shift over the course of conditioning.

We longitudinally tracked individual neurons across training sessions. To identify the same cells across two sessions, we warped the imaging field of view of one day to align with that of another day by applying an affine transformation to the time average of one imaging session; we then applied the same transformation to each region-of-interest (ROI) and located matching ROIs across sessions (Fig. 3a; see Methods).

As mice progressed from naïve to expert levels of behavioral performance, longitudinally tracked cells' activity dramatically increased (see examples in Fig. 3b). Averaged across mice, the majority (69%) of previously unresponsive cells began responding to the stimuli. Meanwhile, cells that were already responsive to the stimulus mostly remained responsive (83%), and only a small fraction (17%) became unresponsive (Fig. 3c, left). For the repeated-exposure mice, on the other hand, tracked cells became less responsive to stimuli, with more than half (55%) of originally responsive cells losing responsiveness, and most unresponsive cells (62%) remaining unresponsive (Fig. 3c, right). The magnitudes of these changes were similar if we considered cells pooled across mice (Supplementary Fig. 3a) rather than averaged. (For an illustration of the turnover of cells of different response-pattern subclasses ("on", "off", "on-off"), see Supplementary Fig. 3b.)

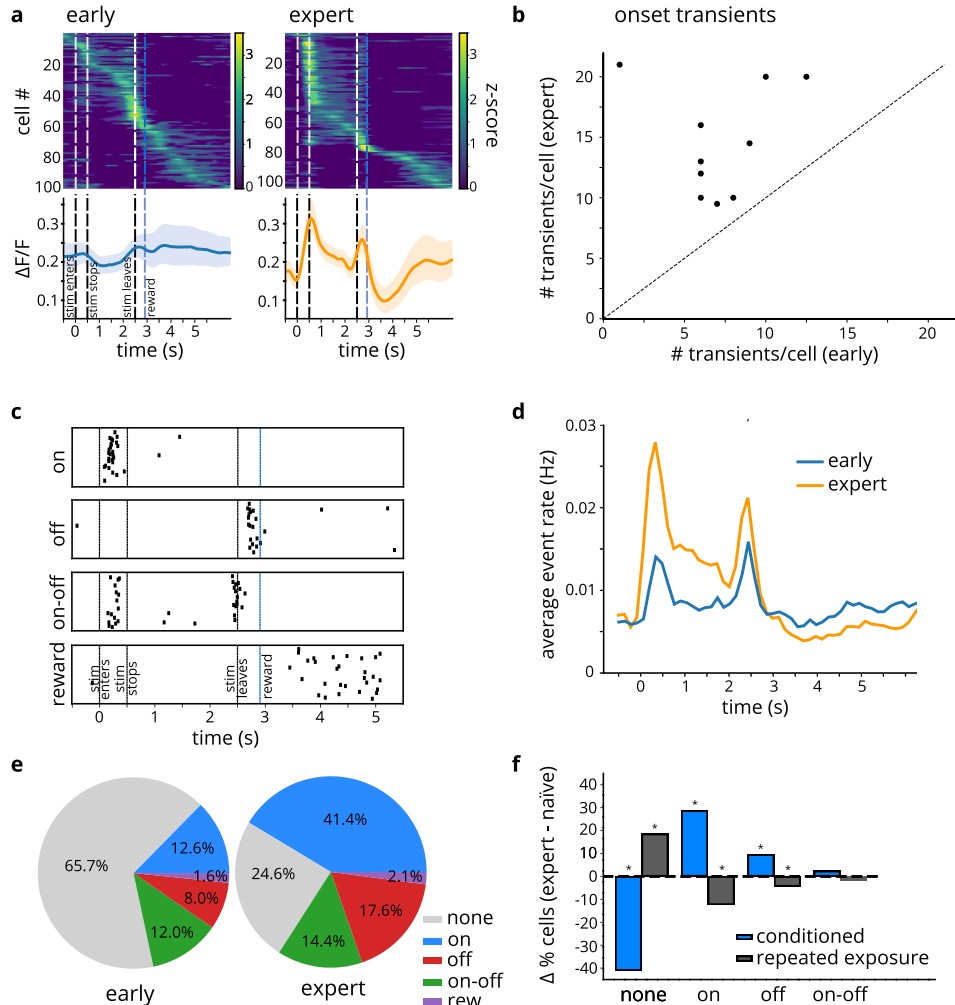

**Fig. 2 | Behavioral training, but not stimulus exposure, increases proportion of stimulus-responsive cells.** (*n* = 10 mice). **a** Cells respond at different times in trial. Top: heat maps from 100 cells in a naïve session (left) and 104 cells in an expert session (right) in an example mouse; Bottom: average across cells for this mouse, for the same early (left) and expert (right) session. Shaded region corresponds to 95% confidence interval. **b** Average number of transients per cell for early and expert sessions (*p* = 0.001, two-sided paired *t*-test). Each point corresponds to 1 mouse. Data shown in Source Data file. **c** Raster plots of calcium transients for 4 example cells show diverse patterns of activity. **d** Transient event rate in naïve (blue) and expert (orange) animals. Rate of transients was calculated across all cells, normalized by number of trials and number of cells. **e** Proportions of different cell types in the population, on early vs expert days for conditioned mice: on cells (blue): $p = 4.33 \times 10^{-54}$; off cells (red): $p = 5.41 \times 10^{-12}$; on-off cells (green): $p = 0.1$; reward cells (purple): $p = 0.4$; none cells (gray): $p = 2.36 \times 10^{-86}$ (based on two-sided Z-test approximation to the binomial, corrected for multiple comparisons; sample sizes: $n_{early} = 1163$ cells; $n_{expert} = 1112$ cells). **f** Change in the proportion of cells of each type (expert - naïve), for conditioned mice that experience reward-pairing (*n* = 10) and for mice that only experience repeated exposure to the stimulus without reward (*n* = 7). For repeated exposure mice: on: $p = 6.77 \times 10^{-8}$; off: $p = 0.03$; on-off: $p = 0.22$; none: $p = 2.6 \times 10^{-12}$ (based on two-sided Z-test approximation to the binomial and corrected for multiple comparisons). Data for conditioned mice same as in **e**. Source data provided in Source Data file.

However, in mice that had experienced reward-pairing, longitudinal cell tracking revealed that cells maintained a "memory" of the task and were relatively stable across consecutive days: 81% of cells were consistently within the same responsive/unresponsive category across consecutive expert sessions. Thus, over the course of learning, individual tracked cells' representations become biased toward reinforced stimuli, but only in conditioned mice. Later, the activity of these same cells stabilized as the animals gained understanding of task rules.

### Training enhances decodability of stimulus and choice

Learning rendered cells not only more responsive, but also more predictive of trial type and the animal's behavior. We trained a support vector machine with a linear kernel to decode the stimulus (the presence or absence of the object) or choice (whether or not the mouse displayed anticipatory licking) from calcium transients in the neuronal population (Fig. 4). Stimulus decoding performance on naïve days was greater than chance but did not reach perfect accuracy, consistent with the unresponsive and unreliable nature of layer 2/3 cells[14–16]. On average, decoding accuracy for both stimulus (Fig. 4a; two-sided paired *t*-test, *p* = 0.007) and choice (Fig. 4b; two-sided paired *t*-test, *p* < 0.001) improved along with animals' performance.

The above analysis used all imaged neurons, including the unresponsive "none" cells. Non-classically responsive cells have previously been shown to be informative in sensory cortex and other cortical regions[43–49]. Interestingly, our unresponsive cells also contributed significantly to the decoders. Unresponsive cells had smaller contributions (decoder weights) in the stimulus classifier than did cells of other response profiles (two-sided Mann–Whitney *U*-test, *p* = 0.002) but had similar weights for choice decoding (two-sided Mann–Whitney *U*-test, *p* = 0.32). Yet even for the stimulus classification, decoding just from "none" cells yielded above chance performance on expert days (76.8% decoder performance; two-sided *t*-test, *p* < 0.001).

Furthermore, even though the decoder could discriminate the trial type classes above chance even in naïve mice, the trial types

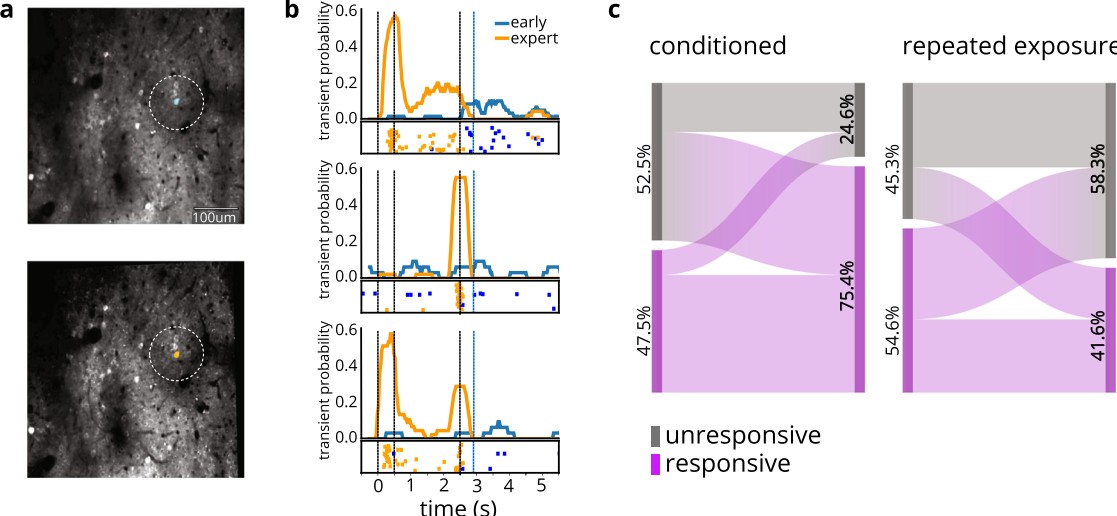

**Fig. 3 | Longitudinally tracked cells' responses exhibit opposing processes of enhancement and habituation in response to conditioning and repeated exposure, respectively. a** Example field of view of example mouse on naïve (top) and expert (bottom) days ($n = 9$ conditioned mice; $n = 7$ repeated exposure mice). A single tracked cell is highlighted. **b** Three example cells' activity on early (blue) vs expert (orange) days. For each cell: PSTH shows probability of calcium transients throughout the timecourse of a "hit" trial; raster plot shows all calcium transients for all "hit" trials in the session. **c** Responsive and unresponsive cells longitudinally tracked across conditioning (left) or repeated exposure without reward (right). Percent cells for responsive and responsive cells was calculated for each mouse and then averaged across mice, to ensure that each mouse's contribution was weighted equally.

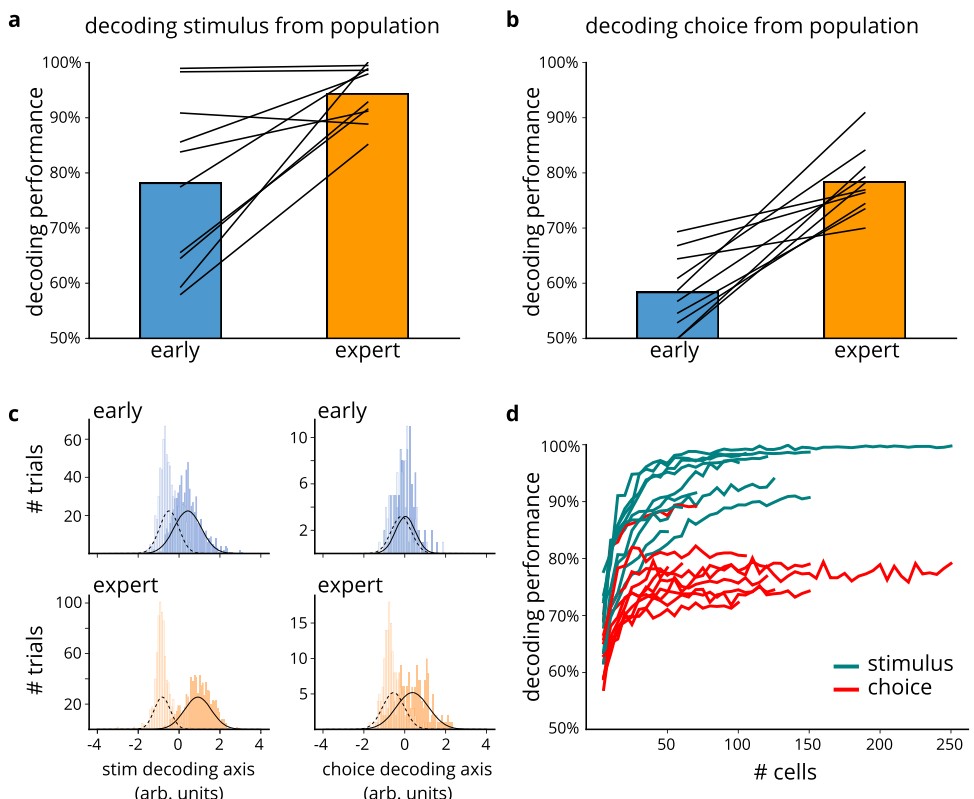

**Fig. 4 | Training enhances decodability of stimulus and choice, from population activity. a** Decoding stimulus from cell population on early (blue) and expert (orange) days ($p = 0.007$, two-sided paired $t$-test). Black lines represent individual mice. Source data shown in Source Data file. **b** Decoding choice from cell population on early (blue) and expert (orange) days ($p < 0.001$, two-sided paired $t$-test). Source data shown in Source Data file. **c** Separability of trial types when decoding from the cell population. Left: stimulus present vs stimulus absent trials for early (top) and expert (bottom) days (filled bars: stimulus present; empty bars: stimulus absent). Right: trials with vs without anticipatory licks for early (top) and expert (bottom) days (filled bars: lick; empty bars: no lick). **d** Decoding performance on expert days for stimulus (turquoise) and choice (red) using increasing number of cells. Curves correspond to individual mice.

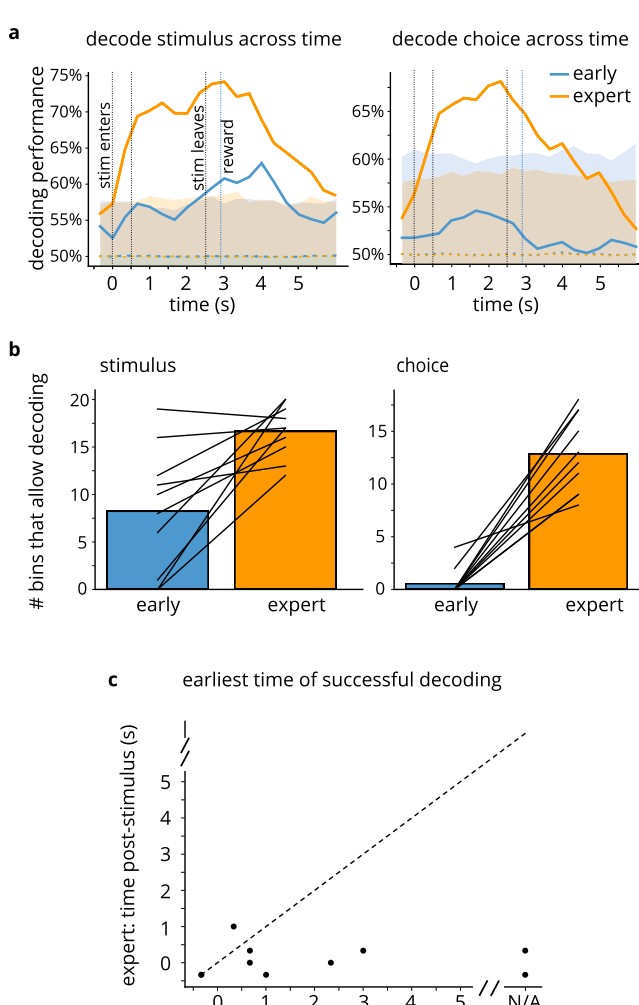

**Fig. 5 | Temporal dynamics of neuronal representations. a** Timecourse illustrating decoding performance, averaged across mice, for stimulus (left) and choice (right) from the population of cells, for real neuronal data (solid lines) and for shuffled data (dashed lines). Blue: early day; orange: expert day. Dashed lines indicate mean of shuffled data; shaded area shows 99% confidence interval for shuffled data (*n* = 10 mice). **b** Number of time bins from which stimulus (left) and choice (right) can be decoded above chance (significantly different from shuffle), on naïve (blue) and expert (orange) days: above-chance decoding performance at more time bins on the trained day (stimulus: *p* = 0.008; choice: *p* = 0.005, two-sided Wilcoxon signed rank test; *n* = 10 mice). Data is provided in Source Data file. **c** First time in trial when stimulus can be decoded: stimulus can be decoded earlier in the trial after training (*p* = 0.03, two-sided Wilcoxon signed rank test). Each dot is one mouse.

became more separable on expert days compared to early-training days (Fig. 4c): projecting neuronal activity across the coding axis (determined by the classifier weights) revealed that the trial type classes diverged on expert days, and therefore that the decoder was more "confident" in classifying the trial types (Supplementary Fig. 4a,b; average earth mover's distance between stimulus classes increased from 0.91 to 1.75; two-sided Wilcoxon signed rank test, *p* = 0.005; average earth mover's distance between choice classes increased from 0.18 to 1.02, *p* = 0.005). Note that for mice in the repeated exposure cohort, the ability to decode the stimulus from neuronal activity did not improve with continued stimulus presentation (Supplementary Fig. 4c), nor did the stimulus classes diverge (Supplementary Fig. 4d).

Since the total number of cells was not identical across different imaging sessions (ranging between 50 and 260 cells; mean = 116 cells), a potential confound could arise if decoding performance scales with

population size[45]. Comparing decoding accuracy using differently sized subsets of the same neuronal population relieved this concern by showing that, while the decoding performance did initially increase with the number of cells used, it rapidly plateaued at sizes smaller than that of our typical imaging datasets (Fig. 4d). In fact, decoding from just 50 cells was above chance (two-sided *t*-test, *p* < 0.001 for both stimulus and choice)−on average 93% for stimulus and 78% for choice. Thus, the different population sizes across sessions cannot explain the learning-dependent improvement in decodability of stimulus and choice.

So far, we have demonstrated that, at least on expert sessions, we were able to decode both stimulus and choice. However, these two measures become increasingly correlated as mouse performance improves: while mice never reach perfect performance, we expect stimulus and choice to be substantially correlated on expert days, when mice mainly choose to lick on stimulus-present trials and inhibit their response on stimulus-absent trials. Therefore, a problem arises: on expert days, cells' activity may truly predict only stimulus *or* choice, and our ability to decode both measures might stem from the correlation between them. To disambiguate our claim from this alternative explanation, we took advantage of the fact that mice do make errors even while performing relatively well, and we re-ran the decoding analysis using trial-balancing[11], whereby each trial type was weighted inversely to the frequency with which it occurred; in other words, rare trial types (miss and false alarm trials) were weighted more strongly than frequent trial types (hit and correct rejection trials). On expert days, when stimulus and choice are correlated, trial-balanced decoding performance for both measures was lower compared to decoding performance using an unbalanced approach (Supplementary Fig. 5a, b; two-sided paired *t*-test, stimulus *p* = 0.03, choice *p* < 0.001), but both remained significantly above chance (Supplementary Fig. 5c; two-sided *t*-test, stimulus *p* < 0.001, choice *p* < 0.001). Consequently, while some degree of decoder performance is attributable to stimulus-choice correlation, the ability to decode one variable is not simply due to its correlation with the other.

Stimulus and choice could also be decoded from individual cells, albeit to a lesser extent than from the entire population. On expert days, the decoder could predict trial type and the animal's response with greater accuracy from a larger percentage of single cells than on early days (Supplementary Fig. 6a). When compared to shuffled data, the stimulus and choice could be decoded above chance from more cells on expert days than on naïve days (Supplementary Fig. 6b; two-sided paired *t*-test, stimulus *p* = 0.004; choice *p* < 0.001). Finally, for individual, longitudinally tracked cells, decoding performance improved as the mice learned the task (Supplementary Fig. 6c; two-sided paired *t*-test, stimulus *p* < 0.001; choice *p* < 0.001). Thus, as mouse behavioral performance improved, individual cells, as well as population activity, became more predictive of trial type and mouse response type.

### Temporal properties of neural representations
We next examined the temporal dimension of the neuronal representations by decoding stimulus and choice at various timepoints throughout the trial. We split the trials into time bins, with each bin containing 10 frames (1/3 of a second), and used the population activity at each time bin to decode stimulus and choice, yielding a decoding timecourse (Fig. 5a). The number of time bins at which the decoder could predict stimulus and choice above chance increased as mice learned the task (Fig. 5b; two-sided Wilcoxon signed rank test, stimulus *p* = 0.008, choice *p* = 0.005). Moreover, the first time bin from which trial type could be predicted shifted earlier over the course of learning (Fig. 5c; two-sided Wilcoxon signed rank test, *p* = 0.03), suggesting that cells' activity encodes information about the stimulus (or future response) earlier in the trial as mice begin to comprehend the task demands[45] and the relevance of the stimulus.

## Population activity encodes time

The above finding implies a temporal aspect to the cells' information coding, but the information encoded still only concerns trial parameters. Consequently, we asked: does neuronal activity also contain information about the progression of time? Specifically, could we predict the position of a given time bin within the temporal sequence from the population activity at that time? Indeed, in expert mice, we were able to decode time from calcium transients, not only at moments when a trial event was occurring (such as the stimulus arriving or leaving, or reward being delivered), but also during intervals when little was changing, such as the 2-s period during which the stimulus was stationary in front of the mouse's face (Fig. 6a, right). In naïve mice (Fig. 6a, left) and in the repeated exposure cohort (Supplementary Fig. 7a), the neuronal activity was less predictive of time bin identity, indicating that the temporal information encoded by the cell population arises due to learning. Varying the bin size (Supplementary Fig. 8a) reveals that the resolution of the encoded temporal information was on the timescale of ~150–200 ms (Supplementary Fig. 8b).

We noticed that training on "hit" trials allowed the decoder to predict time on "hit" trials and "miss" trials, for which the stimulus was present, but not on "correct reject" trials, for which the stimulus was absent. We were unable to decode time on "correct reject" trials even when we trained the decoder on that trial type (Fig. 6a). The inability to predict time on stimulus-absent trials could be due to the lack of tactile signals or lack of animal attention for that trial type.

Interestingly, time coding did not differ between thirsty and satiated states: decoding from the first third versus the last third of a session did not reveal qualitative differences in time coding (Supplementary Fig. 7b). Thus, while time coding may depend on attention, as it arises only in expert mice in a behaviorally relevant context, it also appears to exist independent of motivational state.

Importantly, these timekeeping effects are not inevitable. Decoding time with high precision on a trial-by-trial basis from calcium transients in expert mice means that learning stabilizes previously unstructured trajectories in neuronal state space. Time decoding would be unsuccessful if these trajectories were inconsistent across trials[50] or if they exhibited fixed-point dynamics[38].

Having ascertained that the neuronal population encodes time with remarkable temporal resolution, we naturally asked how: which coding scheme does barrel cortex use to track time? Potential coding schemes include precise time-varying activity patterns of individual cells, stabilization of time constants of ramping activity, or response sequences across the neuronal population akin to those of hippocampal time cells. We found that neuronal activity can be described by the last option: cells are active during temporally-constrained periods that tile the duration of the trial (Fig. 6b; see Methods).

When characterizing the responses of time-tuned cells, we found that these neurons mostly belong to the "responsive" category: 70% of time-tuned cells are responsive in naïve mice, and 89% are responsive in expert mice. In expert mice, time-tuned cells can mostly be classified as "on" cells (46%), though 22% are "off" cells, and 18% are "on-off". In line with this observation, we found that, across the population of cells (Fig. 6c), time-tuned regions cluster around the stimulus onset and offset, consistent with time-tuned cells belonging to "on", "off", and "on-off" classes. Yet, many of these time-tuned regions occur during the intermediate period as well, especially in expert mice (Fig. 6c, bottom). Temporal sequences of neuronal activity constitute an especially effective time coding scheme[36]; the fact that we see evidence of this type of temporal coding in a primary sensory cortical region supports the notion that timekeeping is a distributed process comprising multiple brain networks.

## Movement does not account for learning-dependent changes in activity

Given that our behavioral paradigm is whisker-based, one might speculate that animals' whisking in response to stimulus presentation could modulate barrel cortex responses in a manner that could explain the findings described thus far. To examine this possibility, we analyzed whisker motion from videos of the mice recorded throughout learning (Fig. 7a). We quantified whisking motion as the mean difference between consecutive video frames. We identified bouts of whisking (including whisking during both trials and intertrial intervals) and calculated a whisk-triggered average of neuronal fluorescence (see Methods). Comparison of whisking-driven responses (Fig. 7b, top) and stimulus-driven responses (bottom) of the same population of cells reveals that neurons respond strongly to stimuli, but not to whisking. In addition, the degree of whisking does not increase with learning: if anything, whisking decreases (Fig. 7c).

We also directly tested the contributions of movement to the cells' responses by performing a linear regression, using stimulus onset, stimulus offset, reward, whisking, and licking as features. While the contributions (as determined by coefficients) of stimulus onset and offset increased with learning, the contributions of whisking and licking remained near zero (Supplementary Fig. 9).

Next, we investigated whether movement can explain the aforementioned temporal information encoded by barrel cortex neurons. While the progression of time could be decoded from the activity of the cell population, it could not be decoded from whisker motion (Fig. 7d). The vertical bars in Fig. 7d show that decoders based on whisking consistently misclassify the time within the trial.

Similarly, licking cannot account for the temporal decoding results, as we were able to decode time progression from population activity on "miss" trials during the stimulus (pre-reward) window, when there was no anticipatory licking (Fig. 6a, upper-right panel). In addition, time could not be decoded from licking behavior (Supplementary Fig. 7c, top). Thus, neither of these predominant movements can explain our results.

## Temporal surprise is an effective driver of neuronal activity

Finally, we investigated whether a mismatch of expectation and sensation can modify neuronal activity. Mismatch-related amplification of neuronal activity has been observed in primary visual cortex[26]; in somatosensory cortex, on the other hand, cell responses have been found to be dampened when sensory feedback did not match expectations[27]. These previous studies investigated mismatch through perturbations of optic or tactile flow, by altering the velocity of the visual or tactile stimulus. Sudden shifts in velocity constitute a change of stimulus properties, which might alternatively explain the resulting variations in cortical activity[30].

Given the ability of barrel cortex to represent within-trial time progression as described above, perturbing the timing of the stimulus would be a useful means of disrupting an animal's expectations, without altering the intrinsic properties of the stimulus. Accordingly, we performed a "delayed-offset" experiment in expert mice, in which stimulus offset and corresponding reward were delayed by 1 s on 20% of the trials (Fig. 8a, bottom). We compared trial-averaged fluorescence for delayed-offset vs normal trials and noticed that a large subset of cells (32%, 169/527) exhibited significantly stronger offset responses on the delayed-offset trials compared to normal trials (see Fig. 8a, top and middle for representative example cells). Note that these analyses were corrected for multiple comparisons (false discovery rate). We quantified this difference across the neuronal population and discovered that a majority of cells show a greater response (39% increase in response, on average) to the delayed offset compared to normal stimulus offset (Fig. 8b; two-sided Wilcoxon signed rank test, $p < 0.001$), indicating that surprising events yield stronger signals than expected events. Furthermore, the cell population response to delayed

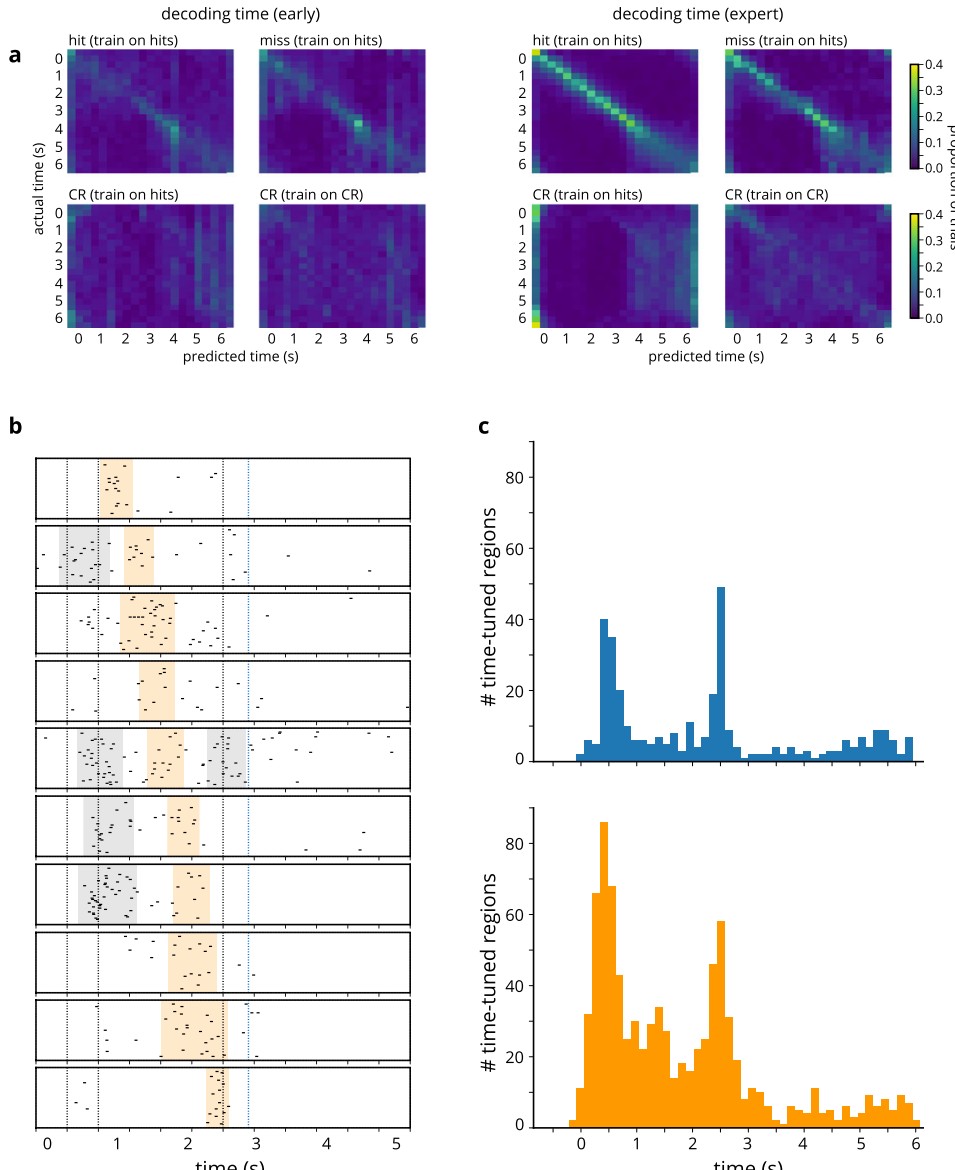

**Fig. 6 | Cell population encodes time with high precision. a** Decoding time from cell population, early (left) and after training (right). In each confusion matrix, columns indicate predicted time; rows indicate actual time. Colors show proportion of trials on which time *y* is predicted to be time *x*. Results were averaged across mice (*n* = 10 mice). Bin size = 10 frames. Sum across row = 1. For each session, time decoding confusion matrices correspond to the following trial types: top left: hit trials, after training the decoder on hit trials; top right: miss trials, after training on hit trials; bottom left: correct rejection (CR) trial, after training on hit trials; bottom right: correct rejection trials, after training on correct reject trials. 4-fold cross-validation was used for all instances of decoding. **b** Cells are tuned to time intervals that span the duration of the trial. Shaded areas correspond to time-tuned regions. Some regions are highlighted (orange) to illustrate how these regions tile the trial timecourse. **c** Histograms indicating the total number of time-tuned regions tuned to different times in the trial, in naïve (blue, top) and expert (orange, bottom) mice.

stimulus offset was on average even stronger (36% increase on average) than the response to stimulus onset (Fig. 8c, *p* < 0.001), whereas for normal trials, the amplitude of the response to stimulus onset exceeded the response to stimulus offset (Supplementary Fig. 10a, *p* < 0.001). Thus, even in a primary sensory area, basic stimulus features are not necessarily the strongest driver of cortical activity.

To ensure that the temporal surprise responses do not merely reflect a change in animals' behavior on the delayed-offset trials, we checked the licking and whisking behaviors on both trial types. Both whisking (Supplementary Fig. 11a) and licking (Supplementary Fig. 11b) behaviors remained similar on normal and delayed-offset trials, indicating that the enhanced neuronal responses on the delayed-offset trials likely stem from the animals' experience of "surprise" or prediction error, rather than altered movements.

Further characterizing the neuronal responses to temporal surprise, we found that "surprise" neurons came from all of the response-pattern categories previously discussed, including stimulus "on" cells (19%), stimulus "off" cells (40%), stimulus "on-off" cells (9%), "reward" cells (8%) and otherwise unresponsive neurons (24%). We also found that found that cells with an "off" response retain an "off" response in the surprising delayed-offset condition, but shift the time of responding to the new offset time during the delayed-offset trials. In addition, some cells with other response patterns, such as "on" cells, are re-recruited and respond once more to the delayed offset (Supplementary Fig. 10b).

Finally, we investigated whether the subset of neurons that preferentially responded to the unexpected stimulus offset in expert mice might have always responded to novelty by checking the responses of these cells on early training days, when all stimuli were novel.

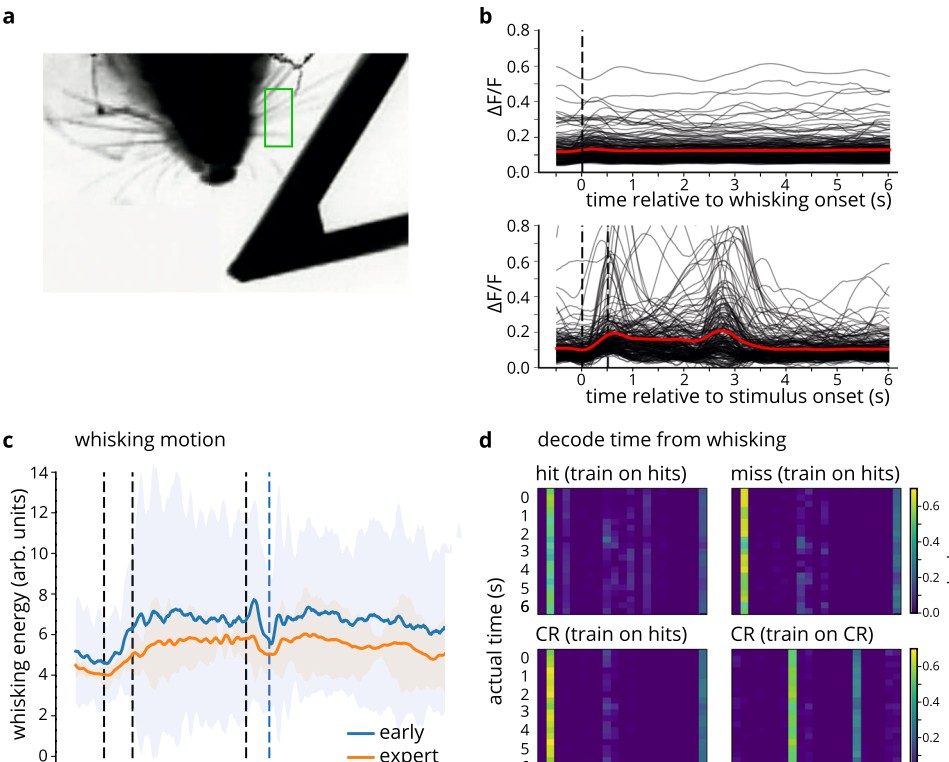

**Fig. 7 | Whisking does not drive cells' responses, and is not responsible for learning-related increases in signal. a** Frame from whisker video: whiskers are imaged from below. Green outline indicates ROI in which whisker motion was analyzed. **b** Top: whisk-triggered average for all cells (*n* = 3 mice); dashed vertical line indicates onset of whisk bout. Bottom: stimulus-driven responses for all cells

(*n* = 210 cells). Red lines show average across cells. **c** Average whisking motion (arbitrary units) on early and expert days (*n* = 3 mice). Solid lines are the mean values; shaded area corresponds to 95% confidence interval. **d** Decoding time from whisking data (same format as Fig. 6a).

Interestingly, we discovered that these "surprise" cells did not respond to the stimulus onset or offset in naïve mice (Supplementary Fig. 10c). On the surface, these observations may seem contradictory. However, the behavioral relevance of the stimulus on naïve and expert days differs markedly: on early-training days, stimuli may be novel, but are still behaviorally meaningless to the mice, and are therefore not necessarily "surprising". From these results, we conclude that unexpected events are even more effective in eliciting layer 2/3 activity than sensory stimuli.

## Discussion
In recent years, a growing body of research has begun to reveal nuances of processing in primary sensory cortices. Contrary to the canonical view of primary sensory cortex as an early stage in a sensory processing hierarchy, we now have reasons to believe that primary cortex may have a role in multisensory integration[4,10,51,52] and in the processing of non-sensory information such as choice, reward, and reward expectation[3,9,11,53]. In addition to this evidence for complexity and higher-level processing in primary cortical areas, parallel evidence contradicts the straightforward notion that primary cortical cells are highly responsive to stimuli of the corresponding modality: for instance, while deep layer cells in barrel cortex respond vigorously to whisker stimulation, even spatially and temporally complex whisker stimuli elicit only low levels of activity in superficial layer cells[14,15], though these cells may respond more to whisker contacts when they are behaviorally relevant[11]. The current study contributes to this alternative view of primary sensory cortex as akin to association cortex, and unveils additional facets of cortical processing.

Here, we demonstrate that learning recruits previously unresponsive cells in barrel cortex, creating a neuronal population that better represents tactile stimuli that have been paired with reward. Previous studies have identified structural and functional plasticity in barrel cortex after sensory deprivation[54,55] as well spine plasticity after learning[23], so one might have anticipated learning to alter neural representations as well. However, there remains disagreement in the field on this issue: some studies have indeed found learning-related functional changes[21], while other studies observed more stability across learning. Kim and colleagues[24] observed a high variability and turnover in the responsiveness and selectivity of barrel cortical cells while mice learned an object-angle discrimination task, but those authors and others[24,25] reported that the proportion of responsive cells remained unchanged across learning. Interestingly, Makino & Komiyama[19] find that the number of layer 2/3 responsive cells actually decreases with learning, though that study is in a different modality, and in an aversive conditioning task, which may explain the discrepancy. In contrast with these latter studies (but in agreement with Chen et al.[21]), we find that the ratio of responsive neurons markedly increases with learning: this recruitment of previously unresponsive cells into a newly responsive neuronal population explains our observation of increased overall touch responsiveness of barrel cortex. As for individual longitudinally tracked cells, we did observe some degree of bidirectional turnover, including both cells that lost responsiveness as well as those that became more responsive. However, training appeared to rebalance these dynamics such that, while only a minority of cells that were originally active lost

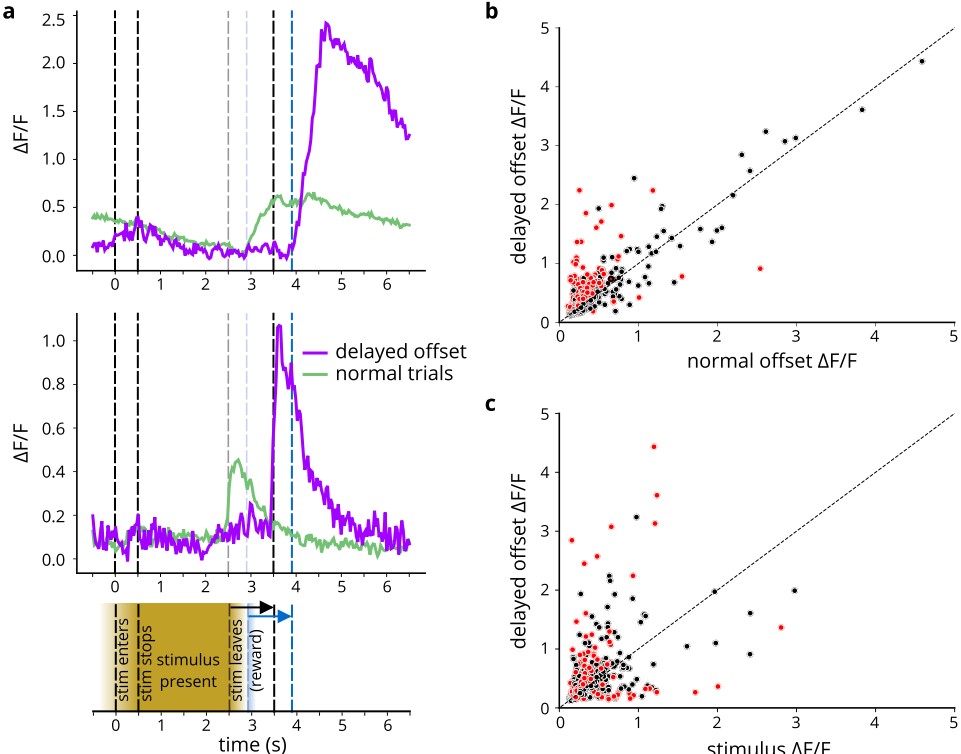

**Fig. 8 | Temporal surprise drives neuronal activity.** $n = 3$ mice (6 sessions). **a** Bottom: schematic of behavior; top: 2 example cells' activity on normal trials (green; 80% of trials) and on delayed offset trials (purple; 20% of trials). **b** Amplitude of late peak for normal vs delayed offset trials for all cells. Red dots indicate cells with significant difference (false detection rate corrected) in their responsiveness, most unresponsive neurons gained stimulus responses.

amplitude between normal and delayed offset trials (normal vs delayed: $p = 1.22 \times 10^{-32}$, two-sided Wilcoxon signed rank test). Source data shown in Source Data file. **c** Amplitude of stimulus response vs offset response on delayed trials ($p = 7.59 \times 10^{-11}$, two-sided Wilcoxon signed rank test). Source data shown in Source Data file.

In stark contrast, repeated exposure to a stimulus without reward-pairing actually reduced barrel cortex responsiveness to the stimulus. The latter phenomenon likely arises due to habituation: whereas mice learn the behavioral relevance of a stimulus followed by reward, they can equally learn the relative unimportance of a stimulus that is presented without positive or negative consequences. Similar findings have been reported in the mouse primary visual cortex[22,56]—repeated presentation of visual gratings led to adaptation of neural responses over the course of behavioral training.

One plausible mechanism for this retuning of sensory representation is contextual information conveyed to barrel cortex via long-range top-down inputs from higher-order cortical areas. Orbitofrontal cortex (OFC) has recently been shown to play an important feedback role, but is unlikely to be involved in the effects described here: input to S1 from OFC has been shown to be activated (and indeed required) in complex tasks such as reversal learning, but OFC only became active during the complex (rule switch) aspects of the task, not during initial learning[57]. However, other top-down inputs, such as those from the prefrontal cortex (PFC), could contribute in our task and others[58–60]. For instance, some high-level information about task context has been identified in both primary auditory cortex (A1) and PFC[59] during auditory behaviors—presumably, this information was conveyed to the primary sensory region from PFC. In addition, prefrontal inputs to primary visual cortex have been shown to enhance mismatch signals[60]. Persuasive evidence also points to top-down modulation of primary sensory cortical activity by retrosplenial cortex during learning[19].

Alternatively or additionally, the effects we observed may have a mechanism of neuromodulatory origin, whereby in the presence of reward, neuromodulatory inputs to barrel cortex create an environment for local plasticity to occur and unveil neuronal responses to the stimuli. While multiple neuromodulatory systems may be implicated, several are particularly compelling. Norepinephrine, for instance, is intimately involved in task engagement and attention and could play a role in the enhancement of sensory responses to behaviorally relevant stimuli[61]. In fact, there exists evidence for the ability of noradrenergic neuromodulation to induce plasticity in primary auditory cortex and strengthen A1 responses to auditory stimuli[62] and to increase neuronal excitability in S1[63]. Acetylcholine, too, has been shown to augment neuronal responses in studies of V1[64,65]. The fact that reward uncoupled with stimulus presentation is sufficient to potentiate cell responses in V1[22] supports the neuromodulation model, as the slow timecourse of neuromodulation may allow for the potentiation of signals even when those signals are not temporally locked to reward. Future studies should investigate this possibility and identify which neuromodulators might be responsible for retuning cortical representations according to reinforcement.

Our results also illustrate how barrel cortex activity becomes more patterned as mice learn the behavioral task, as confirmed by the improved ability of a classifier to decode the presence or absence of the stimulus as well as the nature of animals' responses. Furthermore, learning enabled decoding of the trial type from more time bins and, crucially, earlier in the trial, allowing cells to more quickly access information about the trial. This phenomenon may underlie enhanced reaction times as behaviors become more ingrained, and may be explained by a temporal difference model of reinforcement learning, whereby the reward value is attributed to progressively earlier stimuli

(or perhaps to earlier moments of time within the trial) over the course of learning.

## Barrel cortex encodes time progression

We found that time could be decoded with remarkably high precision from population activity in well-trained mice but not in naïve animals, meaning that cells in barrel cortex could track the progression of time. Previously, in barrel cortex, temporally precise response patterns have been shown to carry sensory information[40], though this finding was obtained in anesthetized rats, making the relevance to behavior and learning somewhat difficult to interpret. Yet, a related finding in the somatosensory cortex of awake primates discovered so-called "memory cells", which displayed specific patterns of activity during the delay period of a tactile working memory task[41]. However, our ability to decode time likely stems from time cell-like responses across the neuronal population rather than from temporally precise activity patterns of individual cells like those observed by Arabzadeh et al.[40]. To our knowledge, our results present the first direct evidence of activity in barrel cortex encoding the progression of time on a behavioral timescale, or of temporal encoding in any rodent primary sensory cortical area.

One could imagine that the temporal information in S1 cells arises due to patterns of whisking: stereotyped whisking patterns would lead to temporally precise series of whisker contacts; consequently, even though the stimulus is not moving, and seemingly no new events occur, a series of stimulus responses could emerge. The timecourse of such signals could conceivably have been the source of the timing information as well as our inability to decode time on stimulus-absent trials. However, we were not able to decode time from whisking (Fig. 7d), ruling out whisker movement as the relevant source of temporal coding in S1.

Another possible source of time coding is the sensitivity of the neuronal population to trial events. After all, time coding arises with learning, as cells gain stimulus responsiveness. We attempted to decode time progression from trial events (stimulus onset, stimulus offset, and reward) as well as animal movements (licking and whisking). Even with all these features included, we were unable to decode time (Supplementary Fig. 7c, bottom), suggesting that the neuronal population is encoding time per se, rather than a series of trial events and movements.

## Cells respond to mismatch of temporal expectations

Our experiments uncovered mismatch signals in barrel cortex: in expert mice, delivering the stimulus offset at an unexpected time yielded the highest levels of cellular activity. Unexpected events have previously been shown to evoke strong responses in primary sensory cortex (e.g., oddball stimuli eliciting responses in "deviance detector" cells of V1[60]). Those kinds of experiments, however, typically use stimuli that have different sensory characteristics from the "non-deviant" stimuli. As a result, the increased responses to oddball stimuli can be explained by synaptic or circuit adaptation to the common stimulus, and a lack of adaptation to the oddball.

Other studies[26,27] have reported altered responses in V1 and S1 to perturbations to optic and tactile flow, respectively, in which sensory input did not match the expected outcome of self-generated motion. Yet, an alternative explanation for these results has been offered[30], invoking cortical cells' preferences for different stimulus velocities, rather than mismatch between expectations and sensory feedback. An important advantage of our experimental design is that we do not change velocity or any other stimulus property to investigate mismatch, but rather the timing of the stimulus. Consequently, the surprising events in our experiment have the same sensory features as expected events, yet they violate the animal's internally constructed model of the world—the temporal structure of the task.

In fact, the "surprise" signals we observe constitute further evidence for the existence and importance of timing information in S1: the enhancement of cells' responses to a disturbance of an event's timing implies that they have "knowledge" of its timecourse, as access to timing information is necessary for sensory cortical cells to exhibit this response to temporal mismatch. The presence of a surprise response in primary sensory cortex may in turn provide the animal with a means for rapid behavioral outcomes to salient, unexpected events, perhaps bypassing slower processing in higher cortical areas more traditionally thought of as associational.

In conclusion, we have demonstrated that primary somatosensory cortex behaves in a more nuanced way than traditionally thought: rather than directly responding to sensory stimuli, sensory cortical representations undergo large bidirectional changes over the course of learning, either expanding to better represent stimuli paired with reward, or habituating and contracting following repeated exposure without reward pairing. Indeed, these effects extend beyond mere stimulus responses, and encompass temporal encoding as well as modeling expected events and their timecourse, implying that timekeeping may be widespread and distributed throughout the brain.

# Methods

## Subjects

All experiments were approved by the Columbia University Institutional Animal Care and Use Committee. 17 C57BL/6J mice (male and female; age range p60–p300) were used in the experiments described here. Mice were housed in groups of 2–5, unless fighting or barbering was observed (in which case they were singly housed). The room in which the animals were housed had a 12 h light/dark cycle: light during the day and dark during the night, switching at 7:00 and 19:00. The ambient temperature was 22 °C. Mice were provided with a running wheel for enrichment as well as *ad libitum* food. During behavioral training, mice were water-restricted and maintained at 80% of their original weight. If daily weighing determined that mice were too light, they were given 5 min of free access to water in their home cage.

## Virus injection and cranial window implant surgeries

CaMKII-GCaMP6f virus (AAV5.CamKII.GCaMP6f.WPRE.SV40, nominal titer $2.3 \times 10^{13}$ gc/mL) was injected into left barrel cortex of the mice. The injection site was targeted at 1.5 mm posterior and 3.5 mm lateral relative to bregma. To accomplish this surgery, mice were anesthetized with isoflurane (3% for induction; 1–2% for maintenance), and subcutaneous analgesics buprenorphine, carprofen, and bupivacaine were administered. Eye ointment was applied, and mice were fixed into the stereotax with earbars. Scalp was shaved and cleaned, and then cut to expose the skull. After locating the injection site, a small area of skull was thinned with a dental drill, until a glass injection pipette could be inserted through cracks in the thinned bone. Virus was injected at depths of 300 μm and 150 μm, with three 50-nL injections at each depth (injections were spaced 1 min apart, with three minutes between depths and before removing the pipette). The pipette was then withdrawn, the thinned area of skull was covered with superglue, and the scalp was sutured closed.

Mice were given an additional carprofen dose 24 h after surgery, and monitored for 5 days. We allowed a three-month interval for the virus to express, and then performed a cranial window implantation surgery.

For this second surgery, intramuscular dexamethasone was administered 3 h prior to surgery. Isoflurane anesthesia was carried out as before, and buprenorphine was administered subcutaneously. Remaining surgical preparation was done as before, but this time a circle of scalp was removed entirely. The skull was cleaned, and a 4-mm diameter craniotomy was made over the barrel cortex. A glass cover slip was placed over the craniotomy and sealed with superglue. Then a metal headplate was affixed to the skull using dental cement.

Subcutaneous buprenorphine was given every 12 h for two days after surgery. Mice were allowed two weeks to recover and for blood to clear from the cranial window.

## Intrinsic signal imaging and 2-photon calcium imaging

Barrel cortex was located using intrinsic signal imaging in anesthetized mice, during which individual whiskers were stimulated with a piezo-electric device at 5 Hz (10 pulses, with a 10-s gap between pulse trains). Under red light illumination, and through a 5× objective, we recorded reflectance at the brain surface with a Rolera-mGi Plus digital camera: changes in blood flow to the barrel corresponding to the deflected whisker yielded a visible change in reflectance.

Two-photon calcium imaging, through a 16×/0.8NA water immersion Nikon objective and with the laser set to 940 nm, was conducted on a Sutter Moveable Objective Microscope; images (512 × 512 pixels) were acquired using ScanImage software with a 30 Hz acquisition frame rate.

Cells in L2/3 of barrel cortex were imaged each day during behavioral training, and the field of view was kept constant between days. The center of the barrel field (C or D row) was targeted for imaging.

## Whisker-based object-detection behavioral paradigm

Our behavioral paradigm was a Pavlovian object-detection task. 10 mice learned the task described below; all data from the "conditioned" cohort come from this group of mice. First, mice were hand-habituated for several days. Second, a "lick-training" phase occurred: mice were water-restricted and head-fixed into the behavioral apparatus, and water was delivered through a lick-port. Initially, water was delivered manually; once mice began to spontaneously lick at the lick-port, water was delivered contingent upon the animals' licking. Licks were measured using an infrared lick-detector: whenever the tongue touched the lick-port, it covered a fiber-optic cable leading to the infrared detector, and the event was registered as a lick.

Finally, the detection task training began. In this paradigm (Fig. 1a), two objects were affixed to opposite arms of a plus-shaped wheel. The other two arms were left empty. A motor rotated the wheel, and brought one of the arms toward the animal's face on each trial. The object "enters" the whisker field at time 0 (Fig. 1a, bottom), and then stops 500 ms later. The wheel remained in this position for 2 s, and then rotated 45 degrees, such that the space between two arms was in front of the face. On each trial, the wheel randomly rotated either clockwise or counter-clockwise. Both directions could result in a rewarded stimulus. The arm that was ultimately presented was also selected at random. If the arm presented to the mouse had an object attached to it, that trial was a "stimulus-present" trial, and was followed by water reward. The onset of water delivery began 407 ms after the wheel began to rotate away (10 ms after the wheel completed the 45 degree rotation). On unrewarded, "stimulus-absent" trials, one of the empty arms was presented, and no water was given. The aforementioned infrared lick-detector was used to measure licks: anticipatory licks were counted during the 2 s leading up to the reward, or lack thereof. Sessions lasted ~30 min, and animals performed on average 150–200 trials per session.

For the *delayed-offset experiment*, stimulus onset did not change, but for a random 20% of trials, the offset was delayed by 1 s (i.e., for those trials, the stimulus (or empty arm) was present for 3 s rather than the usual 2 s). On rewarded trials, the reward timing did not change relative to the offset of the stimulus.

To ensure that any effects we observed were due to stimulus-reward conditioning rather than repeated exposure to stimuli, a separate control group of 7 mice were trained without rewards. These mice had *ad libitum* access to water, and during behavioral training, no water was delivered through the lick-port. The rest of the task parameters remained the same, and these mice were trained, on average, for the same length of time as the mice in the conditioning cohort.

## Histological confirmation of imaging location

After training and imaging experiments were complete, the two-photon laser was used to create a small lesion at the center of the imaging field of view (at a depth of 100–150 μm). Mice were then perfused and their brains harvested. 50 um tangential sections were made, and stained with streptavidin-Alexa 647 in order to visualize the barrels.

## Data processing and analysis

**Motion correction and cell identification.** Suite2p was used for motion correction, to identify ROIs, and to extract their signals[66]. These automatically detected ROIs were then manually curated to remove likely false positives.

**Longitudinal tracking.** Motion-corrected time averages were aligned across any two days by warping one image (time average of Day 2) to match the other (time average of Day 1) with an affine transformation (using the Python OpenCV library). The same transformation was then applied to each ROI mask of Day 2, such that the ROI was warped and shifted to the appropriate location. All ROI locations from the two days were then compared and checked for degree of overlap. Overlapping ROIs were given the same label, which identified them as corresponding to the same cell across the two days. In cases where an ROI from one day overlapped with more than one ROI from the other day, ROI pairs with the greatest degree of overlap were given the same labels. Approximately 10% of ROIs exhibited this kind of "double overlap". We also performed the same longitudinal tracking analysis, excluding ROIs with "double overlap", and saw similar results (Supplementary Fig. 3c).

Once a cell has been found in two imaging sessions and given a label, it maintains that identity when tracked across additional sessions: thus, a cell can be longitudinally tracked across numerous sessions.

**Quantifying cell responses.** Signals were smoothed with a Savitzky–Golay filter. ΔF/F at each timepoint was calculated using a baseline determined by the 8th percentile of a 50 s rolling window centered at that timepoint.

**Transient detection.** For each cell, a threshold was set at 2 standard deviations above the median ΔF/F for that cell. A transient event onset was marked whenever the ΔF/F first crossed the threshold (and the transient event was considered to end when the ΔF/F fell below the threshold) (See Supplementary Fig. 2).

**Classifying cell response types.** Cells were classified as "on", "off", "on-off", "reward", or "none" cells based on the timing of their responses. Once transients were detected for each cell, a histogram was calculated indicating the probability of a transient event occurring at each time within a trial; this probability curve was then compared to that of shuffled data. Shuffled data was generated by doing 5000 iterations of a circular shuffle, whereby the data in each trial was shifted by a random number of frames; thus, any temporal dynamics were preserved, while the relationship of the data to the trial timecourse was disturbed. A Z-test compared the transient probabilities during relevant 1 s intervals (around the stimulus onset and offset, and following reward) to the shuffled data on the same intervals. This test revealed time intervals during which the probability of a transient occurring was greater than chance. If a cell's transient probability was significantly greater than chance only during the stimulus onset bin, that cell was classified as an "on" cell; cells with significant transient probability only at stimulus offset were "off" cells; "on-off" cells had significant transient probability at both intervals, and "reward" cells had significant transient probability after reward. A false discovery rate (FDR) correction accounted for the large number of cells.

**Decoding analyses.** Using Python scikit-learn software[67], we trained linear support vector machine (SVM) classifiers to decode stimulus (i.e., classify trial type as "stimulus-present" or "stimulus-absent"), and to decode choice (i.e., classify the animals' response as "lick" or "no lick"), using 4-fold cross-validation (training on 3/4 of the data and testing on the remaining 1/4). In both cases, we matched the number of trials between the two classes. We computed the separability of the classes by projecting (via dot product) neuronal activity of trials of each class onto the coding direction (as determined by the classifier weights), and quantifying the earth mover's distance between the distributions. Overlap between the distributions corresponds to decoding errors, and the classes are judged to be more separable the further apart the distributions are.

To decode time, we split trials into 10-frame-long time bins (1/3 s duration), and trained SVM decoders to distinguish between time bins. A confusion matrix (see Fig. 6a) with true time bin identity on the *y*-axis and predicted time bins on the *x*-axis illustrated the ability to decode time, by showing the proportion of trials for which a given time bin *y* was predicted to be time bin *x*. The matrix was normalized by the true time bin classes, so the values of each row sum to 1.

**Identifying temporal tuning.** The degree of temporal information in each cell's activity was quantified using the Skaggs spatial information metric[68]. This metric is commonly used to quantify spatial tuning, but has been applied to the temporal dimension as well[69]. We defined cells as time-tuned if their temporal information exceeded the 95th percentile of the temporal information in shuffled data. Among these time-tuned cells, we then identified the specific temporal "regions" to which they were tuned, where each cell's "tuning curve" (probability of transients across time) was above 95% of shuffled curves.

**Whisking analysis**
Throughout conditioning, whisking videos were acquired at 125 Hz using a Sony PS3eye camera. Using the Python OpenCV library, we extracted pixel values within a hand-drawn ROI near the animal's face, on the side of the face contacted by the stimuli. The mean difference between consecutive video frames was used as a measure of whisking motion. Whisking bouts were identified in a similar manner to calcium transient events, by tracking where the whisking motion crossed a threshold. We defined a whisking bout as lasting >0.5 s, so a new bout could not begin until that time interval elapsed. Whisk-triggered averages were computed by aligning fluorescence data with the onset of whisking bouts.

**Statistics and reproducibility**
Sample sizes were chosen to be comparable to sample sizes of other studies in the field, and were not chosen based on statistical methods[20,22,26,48,56]. No data were excluded from the analyses. Both trials and experimental groups were randomized: the trial type was selected randomly for each trial, and animals were allocated to the conditioned and repeated exposure groups at random. However, blinding was impossible, as the cohort identity (conditioned vs repeated exposure) was obvious to both the experimenter and the mice for a number of reasons: repeated exposure mice did not receive water rewards during the task, so the experimenter had to ensure that this cohort of mice received water outside of the task context; in addition, water-restricted mice weighed less.

**Reporting summary**
Further information on research design is available in the Nature Research Reporting Summary linked to this article.

## Data availability
Due to the size of imaging data, data will be available upon request from the authors. Source data are provided with this paper.

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

## Acknowledgements

The authors thank Chris Rodgers for instruction and advice on previous versions of the behavioral apparatus and experiments; Jack Bowler for help with imaging and decoder analyses; Gordon Petty for help with pilot versions of whisking analyses; Ramon Nogueira and Stefano Fusi for advice on decoder analyses; Sam Benezra for advice on imaging; Esther Greeman and Drew Baughman for help with histological sectioning; and Stefano Fusi and Chris Rodgers for comments on the manuscript. Support was provided by NINDS/NIH R01NS094659 and R01NS069679 (RMB), an NSF Graduate Student Research Fellowship (R.J.R.), NINDS/NIH F31NS105490 (D.D.K.), and Kavli Institute for Brain Science (R.J.R., D.D.K.).

## Author contributions

R.J.R. and R.M.B. designed the project. D.D.K. provided extensive assistance in performing surgeries and imaging. R.J.R. performed imaging experiments and analyzed the data. R.J.R. and R.M.B. wrote the manuscript with feedback from D.D.K.

## Competing interests

The authors declare no competing interests.
