## [Peer Review File · Nature Communications]

Learning enhances encoding of time and temporal surprise in
mouse primary sensory cortexREVIEWER COMMENTS

Reviewer #1 (Remarks to the Author):

This is an intriguing study that will be of interest to people working in sensory (particularly tactile) perception, in the neural correlates of goal-directed decision making and in the neural underpinnings of time perception. Its substantial contribution builds on earlier work on population coding of sensory and non-sensory variables in somatosensory cortex, providing suggestive results on how the brain might construct a code for time, using sensory cortex responses as building blocks. The comparison between trained and repeated-exposure animals is illuminating and useful.

Main

My main queries concern the variables that might be driving the time-encoding neurons. I would like the authors to characterize (or control for) these variables more thoroughly. The neurons shown here don't seem to be encoding time in an invariant way, like e.g. some striatal time encoders in the Toso & Diamond paper: instead, the "temporal surprise" neurons vary their temporal tuning according to when events arrive in a trial – thus they are sensitive to the conjunction of events/variables and the time at which those events happen. In other words, they seem to be sensitive to phases in the trial rather than to absolute time.

- Given this joint dependence on events and time, can any events with a tendency to occur at different phases be pulled out more clearly, and the responses explained in terms of specific (sensory or non-sensory) event properties?
- For example, does aligning activity to lick times reveal sensitivity to licking in any subset of time-encoding neurons?
- If present, does any such sensitivity change with learning, possibly explaining how learning enhances coding of time?
- In "temporal surprise" neurons with an enhanced response to a reward delay, can one control (via e.g. cross-correlating with lick rate) for whether the increased response reflects a more urgent attempt to collect the reward once this finally arrives?

The authors go some way towards addressing some of these questions, focusing especially on whisking, but it would be useful to look at a wider set of simultaneously recorded parameters. This is particularly relevant given that the authors' own work (Lacefield et al, Cell Reports; Rodgers et al, Neuron) plus that of multiple other groups (e.g. Yamashita & Petersen, eLife 2016; Bale et al, Curr Biol 2021; etc) shows that learning causes neurons in barrel cortex to become sensitive to licking, as well as water reward and other task variables.

Related to this, when individual tracked neurons' representations became biased towards the reinforced stimulus, did that occur also after controlling for e.g. an increase in licks driven by that stimulus? This could be shown by showing that those neurons did not increase their lick-aligned response, or by checking the bias towards the reinforced stimulus after removing neurons that did increase their lick response.

As mentioned, different phases in a trial typically feature different events (stimulus coming within reach, stimulus immobile, stimulus going out of reach; differences in lick rate; etc). Thus all those recorded parameters, taken together, can provide a (relatively coarse) readout of trial phase, or time. How does that readout compare with what was achieved by temporal decoding from the simultaneously recorded neural populations (Fig. 6)? If the neural population signal was better (more reliable) than that given by trial parameters, then that would help argue that the population was telling time through sequential activation of "time neurons" that become tuned to specific moments in the trial (rather than tuned simply to combinations of sensory and non-sensory parameters).

Minor

The dependence of decoding accuracy on population size and its sensitivity to non-classically

responsive cells in the population has been shown before not just in auditory cortex (as cited) but also (and earlier) in the barrel cortex itself (Safaai et al, J Neurosci 2013). This synergistic noise correlation effect appears also in the visual cortex (Montijn et al, eLife 2015; Osako et al, Curr Biol 2021) and is likely to be a general property of cortical coding (Zylberberg, bioRxiv 2017; Leavitt et al, PNAS 2017; Pruszynski & Zylberberg, Curr Opin Neurobiol 2019), so I would encourage the authors to provide citations across sensory modalities.

Signed - Miguel Maravall

Reviewer #2 (Remarks to the Author):

This manuscript examines learning-dependent changes in barrel cortex. Calcium imaging of the barrel cortex was performed before and after animals were trained on a task in which they received a water reward shortly after the offset of whisker stimulation. The results show a fairly dramatic increase in the number of neurons that respond to whisker stimulation after conditioning (“expert”), and showed that the ability to decode stimuli, choice (lick/no lick), and trial time, increased as a function of training as well. An additional and interesting finding is that there is an increased off-response when the stimulation period is extended from 2 to 3 seconds. Although the authors did attempt to control for the effects of repeated stimulation, some of the conclusions would have been more compelling if the authors had used a standard differential conditioning paradigm with CS+/CS- stimuli to allow for within animal comparisons. The findings are consistent with other studies over the past few decades have reported a wide range of experience-dependent increases in activity in visual, somatosensory, and auditory cortex.

The focus on timing is interesting, although as the authors cite the work of Shuler et al, has shown at the visual cortex also encodes time. In contrast to that work, here the experimental task was not explicitly designed to study the temporal component of the task—e.g., does the increase offset response occur to both short and longer stimuli?

Additionally, the decoding of time is a bit confounded because the task did not have an explicit wait or delay period. Leading to the possibility that anticipatory licking altered whisking or modulated whisker responses contributing to the decoding of time. The authors did attempt to disambiguate the observed effects from potential changes in whisking. But showing that one cannot decode time from raw whisking movement does not really demonstrate that there were not some changes in some whisker motion parameter on early and expert days. Indeed, it seems there may have been a decrease in whisking motion. If whisking inhibits cortical responses the decrease in whisking could contribute to enhanced cortical response. But no statistical was provided for the data in Figure 7c, and I think the SEMs are missing.

Figure 1: It would be helpful to show the licking raster and mean data for a single animal. In the group mean plot (Figure 1b) it was not clear if the SEMs are there but not visible or not provided.

Figure 2c: is the cell in the third row anticipating stimulus offset or do the dots precede stimulus offset because of binning? It seems surprisingly well time-locked for an anticipatory response. Similarly, in figure 8, why does the purple trace go up so sharply before stimulus offset?

Figure 3. It might be helpful to have an additional plot similar to 3C, in which the responsive cells are further subdivided into: on, off, mixed. That is could an on cell become an off cell?

Figure 8. Is there any change in whisking between the 2 and 3 second trials that could potential contribute to the enhanced offset response?

The methods state that the wheel rotated either clockwise or counter-clockwise. Does that mean the direction of whisker stimulation changed across trials? If so, please clarify if there were any differences in responses.

It would be nice to provide some more quantitative data about the quality of the longitudinal tracking. For example, it is stated that in some cases an ROI from one day could overlap with more than one ROI from another day, and in these cases they used the label with the largest overlap. How often was there double ROI overlap? It seems it might be best to discard the cells with significant double ROI overlap.

Perhaps I missed it, but I could not find the time bins used for the decoding in Figure 4. Was it the average activity across the whole trial?

Given that the stimulus decoder is simply detecting stimulation or absence of whisker stimulation were the authors surprised early decoding was not close to 100%, since a single extracellularly recorded neuron can probably have 100% discrimination?

Fig 6A was a bit confusing, it would be helpful to label each panel in Fig 6A as to the training and testing conditions.

What are the units on Figure 7C?

Line 479. Missing superscript citation for Arabzadeh et al.

Supplemental Fig. 6 is too small.

Reviewer #3 (Remarks to the Author):

Learning enhances encoding of time and temporal surprise in primary sensory cortex
Rabinovich, Kato, and Bruno

Summary

The authors investigate the responses of layer 2/3 cells in primary somatosensory cortex over the course of learning a Pavlovian detection task. They found that, contrary to prevailing assumptions, this region is not limited to encoding sensory information, but also shows activity related to surprise, reward, and the temporal structure of a task. They also showed that, while more neurons became active when learning the reward association, neuron activity decreased after a similar amount of exposure with no reward association. Strong evidence is provided to support the main findings that temporal coding and surprise are a strong driver of the activity. Overall, this work opens up new areas of investigation for how sensory processing and time could be interacting during perception. My main comments and concerns are suggestions that would help to improve the clarity and cohesiveness of the findings.

Major Suggestions

- 1) The authors classify neurons as on, on-off, off, reward, etc in Fig 2 as well as responsive and unresponsive in Fig 3. How do these categories relate to the cells that encode time? Are the time cells mixed in with these categories? are they more predominant in one category vs. another? When comparing naive versus expert? Do time cells emerge from certain category of cells?
- 2) It was not clear how many trials the animals performed per session. How does the strength of time encoding progress as a function of time spent performing the task when animals go from thirsty to satiated? Potential observed differences in time encoding would help to provide mechanistic insight into whether these responses reflect learned network dynamics and/or are state-dependent.
- 3) The temporal mismatch signals are interesting, but I'm not exactly sure what to make of it with the current analysis. How does licking behavior change during the delayed offset period? To what extent can this be explained by the "persistence" of the present stimuli vs the "absence" of expected reward? Is it the cases that neurons at the end of the temporal sequences are enhanced and prolong their

firing? or are neurons that that fire at different parts of the sequence "re-recruit" in the extended delay? Sorting temporal sequences on the "non-offset" trials and then seeing how they map onto the "delayed offset" trials and vice versa would help to disentangle this.

Related, It would be helpful to see whether licking behavior changes during surprise trials. If animals begin licking after the normal 2s presentation period or licked more vigorously during these trials, then perhaps some of the differences in fluorescence intensity could be explained in part by behavioral changes.

Minor

- It would be helpful to schematize the trial conditions in Figure, so that it's more clear what Hit, Miss, FA, CR corresponds with respect to this task
- The uses of "sensory cortex" and "somatosensory cortex" seem inconsistent. In some cases, "sensory" looks like a shorthand for "somatosensory" (e.g., Line 90) but other times it is used as a generalization for findings in primary somatosensory cortex and primary visual cortex.
- Line 488 refers to Figure 7c but should refer to 7d for the ability to decode time from whisking data
- Line 624: "were them" should be "were then"

Reviewer #1 (Remarks to the Author):

This is an intriguing study that will be of interest to people working in sensory (particularly tactile) perception, in the neural correlates of goal-directed decision making and in the neural underpinnings of time perception. Its substantial contribution builds on earlier work on population coding of sensory and non-sensory variables in somatosensory cortex, providing suggestive results on how the brain might construct a code for time, using sensory cortex responses as building blocks. The comparison between trained and repeated-exposure animals is illuminating and useful.

Main

My main queries concern the variables that might be driving the time-encoding neurons. I would like the authors to characterize (or control for) these variables more thoroughly. The neurons shown here don't seem to be encoding time in an invariant way, like e.g. some striatal time encoders in the Toso & Diamond paper: instead, the "temporal surprise" neurons vary their temporal tuning according to when events arrive in a trial – thus they are sensitive to the conjunction of events/variables and the time at which those events happen. In other words, they seem to be sensitive to phases in the trial rather than to absolute time.

We followed the suggestions of the reviewer to address this issue (see below).

Note that we do not claim that the time coding cells are "invariant": in fact, the observation that time cannot be decoded on unrewarded trials or in a repeated exposure paradigm suggests that the time coding cells do not behave in a clock-like manner, but depend on various factors, including attention, and as the reviewer points out, surprise.

- Given this joint dependence on events and time, can any events with a tendency to occur at different phases be pulled out more clearly, and the responses explained in terms of specific (sensory or non-sensory) event properties?

We determined the proportions of temporal surprise sensitive neurons that respond to various events. The Results section now reports that surprise neurons were comprised of stimulus on cells (19%), stimulus off cells (40%), stimulus on-off cells (9%), reward cells (8%) and otherwise unresponsive neurons (24%). This diversity of response properties, particularly the fact that a quarter of neurons are unresponsive to the stimulus and reward, indicates the temporal surprise activity is not entirely dependent on any particular response type.

We also have now characterized the trial-related response properties of time-coding cells, as addressed in Reviewer 3's 1st major comment—see below.

- For example, does aligning activity to lick times reveal sensitivity to licking in any subset of time-encoding neurons?

We were completely unable to decode time from licking, suggesting that time coding is not tied to licking behavior. This new analysis is shown in Supplementary Figure 7c (top).

- If present, does any such sensitivity change with learning, possibly explaining how learning enhances coding of time?

Our new Supplementary Figure 9 shows the coefficient values of a multivariate regression on early and expert sessions. Note the near-zero contribution of licking, which does not change with learning.

- In “temporal surprise” neurons with an enhanced response to a reward delay, can one control (via e.g. cross-correlating with lick rate) for whether the increased response reflects a more urgent attempt to collect the reward once this finally arrives?

Addressed by our new Supplementary Figure 11b. We found that licking behavior after stimulus offset is similar between normal and delayed-offset trials, suggesting that increased neuronal response does not result from a change in licking.

The authors go some way towards addressing some of these questions, focusing especially on whisking, but it would be useful to look at a wider set of simultaneously recorded parameters. This is particularly relevant given that the authors’ own work (Lacefield et al, Cell Reports; Rodgers et al, Neuron) plus that of multiple other groups (e.g. Yamashita & Petersen, eLife 2016; Bale et al, Curr Biol 2021; etc) shows that learning causes neurons in barrel cortex to become sensitive to licking, as well as water reward and other task variables.

This is addressed in our above analysis using multivariate regression, shown in Supplementary Figure 9. We found near-zero contributions of motion, including licking, to the regression model even on the expert day.

Related to this, when individual tracked neurons’ representations became biased towards the reinforced stimulus, did that occur also after controlling for e.g. an increase in licks driven by that stimulus? This could be shown by showing that those neurons did not increase their lick-aligned response, or by checking the bias towards the reinforced stimulus after removing neurons that did increase their lick response.

This is addressed in our above analysis using multivariate regression, which factors out any effect of licking in assessing the stimulus effects.

As mentioned, different phases in a trial typically feature different events (stimulus coming within reach, stimulus immobile, stimulus going out of reach; differences in lick rate; etc). Thus all those recorded parameters, taken together, can provide a (relatively coarse) readout of trial phase, or time. How does that readout compare with what was achieved by temporal decoding from the simultaneously recorded neural populations (Fig. 6)? If the neural population signal was better (more reliable) than that given by trial parameters, then that would help argue that the population was telling time through sequential activation of “time neurons” that become tuned to specific moments in the trial (rather than tuned simply to combinations of sensory and non-sensory parameters).

We have checked temporal resolution used by the time decoding analysis: we decoded time using different numbers of time bins, and examined the difference between neighboring bins. Specifically, we compared the diagonal of the confusion matrix (where the decoder correctly classified the time bin) with the neighboring (off-diagonal) bins. The result is plotted in Supplementary Figure 8. We found that we were able to distinguish neighboring time bins when the bins were as small as ~150ms. Time decoding with high temporal resolution (e.g., 150-msec bins) suggests that we can accurately decode the passage of time even within a trial phase, such as the stationary period of the object presentation.

However, we also did perform the suggested analysis (decoding time progression from trial events): we found that decoding from trial events (stimulus onset, offset, and reward), plus whisking and licking, yields lower decoding performance compared to our reported decoding from population activity. Supplemental Figure 7c (bottom) shows the confusion matrix of this new analysis: from this plot, it is clear that the diagonal indicative of time coding is missing.

Minor

The dependence of decoding accuracy on population size and its sensitivity to non-classically responsive cells in the population has been shown before not just in auditory cortex (as cited) but also (and earlier) in the barrel cortex itself (Safaai et al, J Neurosci 2013). This synergistic noise correlation effect appears also in the visual cortex (Montijn et al, eLife 2015; Osako et al, Curr Biol 2021) and is likely to be a general property of cortical coding (Zylberberg, bioRxiv 2017; Leavitt et al, PNAS 2017; Pruszynski & Zylberberg, Curr Opin Neurobiol 2019), so I would encourage the authors to provide citations across sensory modalities.

We have now added citations across sensory modalities.

Reviewer #2 (Remarks to the Author):

This manuscript examines learning-dependent changes in barrel cortex. Calcium imaging of the barrel cortex was performed before and after animals were trained on a task in which they received a water reward shortly after the offset of whisker stimulation. The results show a fairly dramatic increase in the number of neurons that respond to whisker stimulation after conditioning (“expert”), and showed that the ability to decode stimuli, choice (lick/no lick), and trial time, increased as a function of training as well. An additional and interesting finding is that there is an increased off-response when the stimulation period is extended from 2 to 3 seconds. Although the authors did attempt to control for the effects of repeated stimulation, some of the conclusions would have been more compelling if the authors had used a standard differential conditioning paradigm with CS+/CS- stimuli to allow for within animal comparisons. The findings are consistent with other studies over the past few decades have reported a wide range of experience-dependent increases in activity in visual, somatosensory, and auditory cortex.

While an unrewarded stimulus would have been an interesting comparison within this task, the simpler conditioning paradigm was sufficient to address our main questions about temporal encoding and temporal surprise. Additionally, a CS- within a conditioning task is not necessarily equivalent to repeated exposure. For instance, enhancement of CS- responses has been noted in some experiments (Benezra et al., 2021, BioRxiv; Poort et al., 2015 Neuron). These second-order questions would make for useful follow-ups.

The focus on timing is interesting, although as the authors cite the work of Shuler et al, has shown at the visual cortex also encodes time. In contrast to that work, here the experimental task was not explicitly designed to study the temporal component of the task—e.g., does the increase offset response occur to both short and longer stimuli?

We would have described the work of Shuler slightly differently.

While Shuler and Bear (2006) did study timing responses of neuronal activity, they did not show that the cells encode the progression of time (nor related measures such as event duration). Rather, the focus of their study was about the emergence of *reward-timing* in Layer 4 neurons in visual cortex. They found, with learning, cells began to exhibit ramping responses that peaked at the time of the reward presentation, but they did not attempt to decode time from the cells' responses. Nor do they describe "tiling" of response times of the cell population, which is what we observe.

To address the reviewer's concern, we now state the following in reference to Shuler's work: "Some of the aforementioned studies that found learning-related influences on V1 activity also noted that cells in V1 anticipated the timing of reward and punishment. Yet, no study has directly investigated time coding in sensory cortex."

Additionally, the decoding of time is a bit confounded because the task did not have an explicit wait or delay period. Leading to the possibility that anticipatory licking altered whisking or modulated whisker responses contributing to the decoding of time. The authors did attempt to disambiguate the observed effects from potential changes in whisking. But showing that one cannot decode time from raw whisking movement does not really demonstrate that there were not some changes in some whisker motion parameter on early and expert days. Indeed, it seems there may have been a decrease in whisking motion. If whisking inhibits cortical responses the decrease in whisking could contribute to enhanced cortical response. But no statistical was provided for the data in Figure 7c, and I think the SEMs are missing.

The regression analyses we performed for Reviewer 1 largely address this concern of Reviewer 2. As shown Supplementary Figure 9, the coefficients for whisking are near 0 on average and skewed slightly positive. This is consistent with other studies (e.g., Rodgers et al., 2021; Gentet et al., 2012), which find either little or no effect of whisking on the spiking of layer 2/3 cells. An exception is inhibitory interneurons of the somatostatin type (Gentet et al., 2012), but these do not contribute substantially to our data.

We have added a shaded area corresponding to a 95% confidence interval to Figure 7c. As can be seen, there is no obvious statistical difference between the early and expert whisking.

Figure 1: It would be helpful to show the licking raster and mean data for a single animal. In the group mean plot (Figure 1b) it was not clear if the SEMs are there but not visible or not provided.

SEMs are present, but very small, such that they are not visible at the scale of the subpanel—we now explicitly note this in the legend. We have added the licking data for two example mice (with shaded SEMs), now shown in Supplementary Figure 1.

Figure 2c: is the cell in the third row anticipating stimulus offset or do the dots precede stimulus offset because of binning? It seems surprisingly well time-locked for an anticipatory response. Similarly, in figure 8, why does the purple trace go up so sharply before stimulus offset?

The second and third rows in Fig 2c are binned/smoothed the same way, so the difference in transient times between the two cells must be meaningful rather than an artifact. We suspected that this cell (third row) was an offset-anticipating cell, but cannot make this claim conclusively.

However, smoothing does have an effect on the apparent timing of cell responses in Figure 8. We have replaced the smoothed traces with the raw data in Figure 8a, and have virtually eliminated the pre-offset rise. (Note this should not affect the timing of the transients in Fig 2c, because the smoothed and raw traces “cross” during the rise time, and thus both reach the threshold for transient detection at around the same time.)

Figure 3. It might be helpful to have an additional plot similar to 3C, in which the responsive cells are further subdivided into: on, off, mixed. That is could an on cell become an off cell?

This new plot is now shown in Supplemental Figure 3b.

Figure 8. Is there any change in whisking between the 2 and 3 second trials that could potential contribute to the enhanced offset response?

Addressed by our new Supplementary Figure 11a. We found that whisking patterns after stimulus offset were similar between normal (2 second) and delayed-offset (3 second) trials, mainly characterized by a low amplitude dip in whisking at the time of stimulus offset.

The methods state that the wheel rotated either clockwise or counter-clockwise. Does that mean the direction of whisker stimulation changed across trials? If so, please clarify if there were any differences in responses.

The rotation direction of the wheel was randomly selected on each trial, so the direction does change across trials. However, both directions co-occur with reward, so the direction of stimulation would not affect our findings. We now clarify this point in the Methods section.

Had we collected rotation-direction data, separating clockwise vs counterclockwise trials would indeed have been interesting to check for any differences in cell responses to the different directions: for instance, maybe some unresponsive cells may have actually been responsive to only one direction, but

those responses are hidden by the fact that we do not distinguish the rotation directions. Thus, while our results do not demand this analysis, it might have added an additional dimension to our story and may be included in future studies.

It would be nice to provide some more quantitative data about the quality of the longitudinal tracking. For example, it is stated that in some cases an ROI from one day could overlap with more than one ROI from another day, and in these cases they used the label with the largest overlap. How often was there double ROI overlap? It seems it might be best to discard the cells with significant double ROI overlap.

~10% of ROIs had double overlap. We have now tried excluding them as suggested, and saw similar results as before. This new analysis and the results with exclusion are now reported in the Methods and in Supplemental Figure 3c.

Perhaps I missed it, but I could not find the time bins used for the decoding in Figure 4. Was it the average activity across the whole trial?

In the “*Decoding analyses*” portion of the Methods section, we mention that we split trials into 10-frame-long time bins (1/3 second duration).

Given that the stimulus decoder is simply detecting stimulation or absence of whisker stimulation were the authors surprised early decoding was not close to 100%, since a single extracellularly recorded neuron can probably have 100% discrimination?

As we now note in the relevant Results section, we would not necessarily expect 100% decoding performance for a single extracellularly recorded neuron in superficial layers: L2/3 cells are only sparsely active (Ramirez et al., 2014; Estebanez et al., 2012; Barth & Poulet, 2012). Even our most responsive cells do not respond on every trial. This is different from an extracellularly recorded L4 or L5/6 cell.

Additionally, our stimulus was a relatively weak one compared to standard laboratory stimuli (such as piezo flick), which could in part explain <100% decoding performance. Nonetheless, even for our stimuli, the stimulus decoding performance in the expert mouse was ~94%.

Fig 6A was a bit confusing, it would be helpful to label each panel in Fig 6A as to the training and testing conditions.

We have now labeled each panel, as suggested.

What are the units on Figure 7C?

We have clarified the units in the axis label (arbitrary units of motion energy).

Line 479. Missing superscript citation for Arabzadeh et al.

Superscript citation added.

Supplemental Fig. 6 is too small.

This is now Supp Fig 7, and has been altered with additional subpanels.

Reviewer #3 (Remarks to the Author):

Learning enhances encoding of time and temporal surprise in primary sensory cortex
Rabinovich, Kato, and Bruno

Summary

The authors investigate the responses of layer 2/3 cells in primary somatosensory cortex over the course of learning a Pavlovian detection task. They found that, contrary to prevailing assumptions, this region is not limited to encoding sensory information, but also shows activity related to surprise, reward, and the temporal structure of a task. They also showed that, while more neurons became active when learning the reward association, neuron activity decreased after a similar amount of exposure with no reward association. Strong evidence is provided to support the main findings that temporal coding and surprise are a strong driver of the activity. Overall, this work opens up new areas of investigation for how sensory processing and time could be interacting during perception. My main comments and concerns are suggestions that would help to improve the clarity and cohesiveness of the findings.

Major Suggestions

1) The authors classify neurons as on, on-off, off, reward, etc in Fig 2 as well as responsive and unresponsive in Fig 3. How do these categories relate to the cells that encode time? Are the time cells mixed in with these categories? are they more predominant in one category vs. another? When comparing naive versus expert? Do time cells emerge from certain category of cells?

We have now performed this analysis and discuss the following in the Results section: time cells are mostly “responsive” cells; 70% of time cells are responsive in naïve mice, and 89% are responsive in expert mice. In expert mice, time cells are mostly classified as “on” cells (46%), though 22% are “off” cells, and 18% are “on-off”. This is in line with the results illustrated in the histograms in Figure 6c, which show the temporal location of time-tuned regions: time-tuned regions cluster around stimulus onset and offset, suggesting that those tuned regions belong to on, off, and on-off cells. In expert mice, time-tuned regions in the intermediate period between stimulus onset and offset likely belong to either “unresponsive”/none cells, or to cells that also have an “on” or “off” response (such as the cells shown in the 5th, 6th, and 7th raster plots of Figure 6b).

2) It was not clear how many trials the animals performed per session.

Imaging sessions lasted ~30 minutes, and animals performed an average of 150-200 trials per session. We now note this in the Methods.

How does the strength of time encoding progress as a function of time spent performing the task when animals go from thirsty to satiated? Potential observed differences in time encoding would help to provide mechanistic insight into whether these responses reflect learned network dynamics and/or are state-dependent.

We followed this suggestion and decoded time from different blocks of trials: decoding from the first third versus last third of the session did not reveal qualitative differences in time coding. This new analysis is shown in Supplementary Figure 7b. Thus, time coding likely does reflect learned network dynamics, and is independent of internal motivational state.

3) The temporal mismatch signals are interesting, but I'm not exactly sure what to make of it with the current analysis. How does licking behavior change during the delayed offset period?

Addressed by our new Supplementary Figure 11b. We found that licking behavior after stimulus offset is similar between normal and delayed-offset trials.

To what extent can this be explained by the "persistence" of the present stimuli vs the "absence" of expected reward?

It is true that we cannot entirely disambiguate whether the "surprise" responses are due to the unexpected continued presence of the stimulus versus the surprising absence of reward. Perhaps different cells' responses can be explained by one or the other option. However, the increased response tends to coincide with stimulus offset, rather than ramping with continued stimulus presence, which leads us to believe that the response enhancement is due to the timing of the offset. Of course, the delayed offset and persistence of the present stimulus are inextricably linked, but the offset is a temporally defined event, while persistence of the stimulus is more drawn out. In our response to the next suggestion, we further characterize the cells' responses on delayed-offset trials.

Future studies could further investigate mismatch responses under a variety of conditions and may help disambiguate more subtle aspects of temporal surprise responses.

Is it the cases that neurons at the end of the temporal sequences are enhanced and prolong their firing? or are neurons that that fire at different parts of the sequence "re-recruit" in the extended delay? Sorting temporal sequences on the "non-offset" trials and then seeing how they map onto the "delayed offset" trials and vice versa would help to disentangle this.

We agree this is an interesting question. We performed the suggested analysis (see new panel (b) added to Supplementary Figure 10); we found that mostly, cells with an "off" response retain an "off" response, but shift the time of responding to the new offset time during the delayed-offset trials, rather than prolonging their responses. In addition, some cells with other response patterns, such as "on" cells, are re-recruited and respond once more to the delayed offset.

Related, It would be helpful to see whether licking behavior changes during surprise trials. If animals begin licking after the normal 2s presentation period or licked more vigorously during these trials, then perhaps some of the differences in fluorescence intensity could be explained in part by behavioral changes.

We now report that, licking behavior after stimulus offset is similar between normal and delayed-offset trials (see new Supplementary Figure 11b).

Minor

- It would be helpful to schematize the trial conditions in Figure, so that it's more clear what Hit, Miss, FA, CR corresponds with respect to this task

We have now labeled each panel, as suggested.

- The uses of “sensory cortex” and “somatosensory cortex” seem inconsistent. In some cases, “sensory” looks like a shorthand for “somatosensory” (e.g., Line 90) but other times it is used as a generalization for findings in primary somatosensory cortex and primary visual cortex.

Our intent has been to always use “sensory cortex” as a generalization, to refer to somatosensory/visual/auditory cortices (though a particular example could come from any one region). This is in fact how we meant to use the phrase in line 90 as well: we moved on from a discussion of striatum, amygdala, and association cortex to a discussion of sensory cortex, of which barrel cortex is an instantiation. We have rephrased to clarify.

- Line 488 refers to Figure 7c but should refer to 7d for the ability to decode time from whisking data

Typo fixed.

- Line 624: “were them” should be “were then”

Typo fixed.

REVIEWER COMMENTS

Reviewer #1 (Remarks to the Author):

The authors have addressed my comments. The added analyses in the new supplementary figures help clarify the results. I support publication.

Supp. Fig. 8a has a typo: I believe the bin size on the right is 70 ms, not 100 ms.

Reviewer #2 (Remarks to the Author):

The authors have addressed my concerns, and I think this paper provides a valuable contribution to understanding primary sensory plasticity and contributions to expectation.

Relating to previous studies on timing in primary cortex, I would also point to Chubykin, ... Shuler (2013) for very compelling evidence for V1 in tracking time.

Reviewer #3 (Remarks to the Author):

the authors have addressed all my concerns.

REVIEWERS' COMMENTS

Reviewer #1 (Remarks to the Author):

The authors have addressed my comments. The added analyses in the new supplementary figures help clarify the results. I support publication.

Supp. Fig. 8a has a typo: I believe the bin size on the right is 70 ms, not 100 ms.

Fixed.

Reviewer #2 (Remarks to the Author):

The authors have addressed my concerns, and I think this paper provides a valuable contribution to understanding primary sensory plasticity and contributions to expectation.

Relating to previous studies on timing in primary cortex, I would also point to Chubykin, ... Shuler (2013) for very compelling evidence for V1 in tracking time.

We have added this reference.

Reviewer #3 (Remarks to the Author):

the authors have addressed all my concerns.